# High radiative forcing climate scenario relevance analyzed with a ten-million-member ensemble

Marcus C. Sarofim [1] ✉, Christopher J. Smith[2,3,4], Parker Malek[5], Erin E. McDuffie [1], Corinne A. Hartin [1], Claire R. Lay[5] & Sarah McGrath[5]

Developing future climate projections begins with choosing future emissions scenarios. While scenarios are often based on storylines, here instead we produce a probabilistic multi-million-member ensemble of radiative forcing trajectories to assess the relevance of future forcing thresholds. We coupled a probabilistic database of future greenhouse gas emission scenarios with a probabilistically calibrated reduced complexity climate model. In 2100, we project median forcings of 5.1 watt per square meters (5th to 95th percentiles of 3.3 to 7.1), with roughly 0.5% probability of exceeding 8.5 watt per square meters, and a 1% probability of being lower than 2.6 watt per square meters. Although the probability of 8.5 watt per square meters scenarios is low, our results support their continued utility for calibrating damage functions, characterizing climate in the 22nd century (the probability of exceeding 8.5 watt per square meters increases to about 7% by 2150), and assessing low-probability/high-impact futures.

The warmest standard future scenarios used as inputs for both the fifth and sixth Coupled Model Intercomparison Projects (CMIP5 and CMIP6) reach a forcing of 8.5 watts per square meters ($W\,m^{-2}$) above pre-industrial levels by 2100: namely representation concentration pathway (RCP) 8.5 for CMIP5, and shared socioeconomic pathway-representative concentration pathway (SSP-RCP) SSP5-8.5 for CMIP6[1–5].

RCP8.5 has been characterized as a scenario that assumes high population growth, low-income growth, and modest improvements in energy intensity over the next century, leading to high energy demand and greenhouse gas (GHG) emissions[1,6]. The research community originally developed RCP8.5 as a "plausible future" representing the upper 10th percentile of radiative forcing from scenarios at the time of development in 2007[1]. However, this is not a rigorous method for determining the probability of exceedance of a scenario. Because the underlying scenarios were not designed to be equally likely, the fact that 8.5 $W/m^2$ represented the 10th percentile of available scenarios does not imply an assessment that there was a 10% likelihood of exceeding 8.5 $W/m^2$. Pedersen et al.[7] discuss the history of published

critiques of the lack of probability-based scenario designs in the IPCC process. Meanwhile, SSP5-8.5 followed a different socioeconomic path relative to RCP8.5, with low population growth coupled with high-income growth that drives energy demand, demonstrating that the same forcing scenario can be reached by very different pathways[3,5].

Due to the heavy use of RCP8.5 and SSP5-8.5 by scientific publications and assessments, questions regarding the plausibility[8–11] or implausibility[12–16] of 8.5 $W\,m^{-2}$ scenarios have steadily increased. Additionally, the IPCC AR6 report stated that "the likelihood of high-emissions scenarios such as RCP8.5 or SSP5-8.5 is considered low in light of recent developments in the energy sector"[17]. Due to these discussions in the literature, the recent ScenarioMIP workshop report stated an intent to use a high-emission scenario for the seventh CMIP assessment (CMIP7) that would be "likely below SSP5-8.5" and "possibly near 7 $W\,m^{-2}$"[2].

Importantly, while most of these evaluations of RCP8.5 or SSP5-8.5 have focused on the plausibility or implausibility of the fossil fuel usage and/or emissions pathways in the scenario, it is important to

[1]US Environmental Protection Agency, 1200 Pennsylvania Ave NW, Washington, DC 20460, USA. [2]Met Office Hadley Centre, Exeter EX1 3PB, UK. [3]School of Earth and Environment, University of Leeds, Leeds LS2 9JT, UK. [4]International Institute for Applied Systems Analysis (IIASA), 2361 Laxenburg, Austria. [5]Abt Global Inc., Tabor Center, 1200 17th Street, 10th Floor, Denver, CO 80202, USA. ✉e-mail: Sarofim.marcus@epa.gov

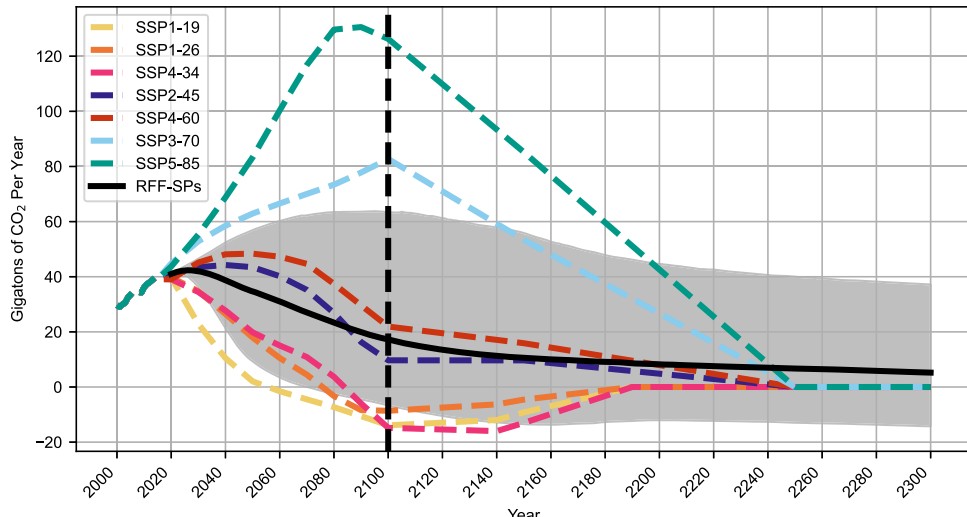

**Fig. 1 | Net annual emissions of CO₂ from RFF-SPs and SSPs.** The black line is the mean and the gray band is the 95th and 5th percentiles for CO₂ (carbon dioxide) emissions from the Resources for the Future Socioeconomic Projections (RFF-SP) scenarios. Colored lines represent CO₂ emissions from six shared socioeconomic pathway-representation concentration pathways (SSP-RCPs), from 2000–2300 from Meinshausen et al. (2020). Source data are provided as a Source Data file.

recognize that when these scenarios are used as input to global climate models (GCMs), they are primarily used as concentration pathways rather than emission scenarios. Several studies discuss the potential importance of carbon cycle uncertainties, though without analyzing the probabilistic implications of including those uncertainties[16,17]. Huard et al.[18] have also highlighted this issue of evaluating scenarios based on their emissions versus their concentrations. Radiative forcing is used as a simplifying aggregate metric rather than comparing individual greenhouse gas concentrations.

In order to address these issues of lack of probabilistic assessment of scenarios, and the evaluation of most of the literature on emissions rather than forcing exceedances, this paper answers two related questions: (1) given a probabilistic ensemble of socioeconomic emissions, what is the likelihood of exceeding 8.5 W m⁻² by the end of the century and (2) what is the appropriate forcing to use for a high-end scenario in climate analysis?

To answer these questions, this paper leverages some other research. First, we use an ensemble of probabilistic emission projections based on expert elicitation of future rates of change of country-level population growth, economic growth, and emissions intensity (the resources for the future socioeconomic projections, or RFF-SPs[19,20]) augmented with improved methods to account for emissions of aerosols and minor GHGs[21], and couple these with a simple climate model, FaIR version 2.1[22], to generate a multi-million-member ensemble to derive a probability density function for end of century radiative forcing. Second, we use the forcing probability density function from this ensemble of equally probable emissions scenarios and climate parameter combinations to determine the likelihood of exceeding 8.5 W m⁻² in 2100, 2150, and 2300. We show that the probability of exceeding 8.5 W m⁻² by 2100 is 0.5%, but we also discuss considerations that, for several applications, such a scenario continues to be useful. We recognize the challenges of accurately capturing the probability of rare events that might have substantial consequences for future emissions and concentration changes, but quantitative analysis of these probabilities has value despite the underlying limitations.

## Results
### Analysis of emissions exceedance probabilities
The RFF-SP scenarios were developed by resources for the future to provide socioeconomic and emissions projections in a probabilistic format through the year 2300. Comparing emission levels across the

RFF-SP scenarios to those in the seven most widely used SSP-RCP scenarios (SSP1-1.9, SSP1-2.6, SSP4-3.4, SSP2-4.5, SSP4-6.0, SSP3-7.0, and SSP5-8.5) (Fig. 1) shows that the RFF-SP upper 95% emissions scenarios are 53 Gigatons of CO₂ (Gt CO₂) below emissions from SSP5-8.5 in 2100, and also 9 Gt CO₂ below SSP3-7.0. Meanwhile, the fifth percentile of the RFF-SP emissions remains above the SSP1-1.9 scenario through 2100. However, looking past 2100, the RFF-SPs include a substantial probability of above zero emissions through 2300, whereas all the SSP-RCPs are constrained to reach zero CO₂ emissions in 2250. Table 1 provides numerical comparisons in the year 2100. The 2100 emissions (both CO₂ and CO₂ equivalents or CO₂e) and the cumulative CO₂ emissions from 2020 to 2100 for each SSP-RCP scenario are included in the left column for comparison. Columns 2–4 show the probability that emissions from the RFF-SP scenarios exceed those of the SSP-RCP scenario for CO₂, CO₂e, and cumulative carbon, respectively. Table 1 shows that the probability of the emissions from the RFF-SPs exceeding those from SSP5-8.5 in 2100 are substantially below 1% by any metric. For SSP3-7.0, that probability is about 1%. The median RFF-SP scenario (17.3 Gt CO₂, 30 Gt CO₂e, 2490 Gt C) lies between SSP2-4.5 and SSP4-6.0 for end-of-century emissions, and just below SSP2-4.5 for cumulative carbon emissions. The fact that the SSP1 family of scenarios and the SSP4 family of scenarios are developed by different modeling groups with different assumptions about underlying economic and technological drivers explains how the emissions from the higher SSP4-3.4 scenario can drop below the emissions from the lower SSP1-1.9 and SSP1-2.6 scenarios by 2100, recognizing that the latter two scenarios still have a lower radiative forcing in 2100.

### Analysis of radiative forcing exceedance probabilities
Figure 2 shows the radiative forcing from the SSP-RCP scenarios (run in FaIR with the same calibration) overlaid on top of the 90% bounds of this analysis. Here, the 95th percentile of radiative forcing from the RFF-SPs is still substantially below SSP5-8.5 in 2100, but comparable to SSP3-7.0 (in contrast to the emissions). Even though the SSP-RCP fossil fuel emissions are constrained to reach zero in 2250, the radiative forcing in SSP5-8.5 remains above the 95th percentile from the RFF-SPs, though the 95th percentile of the RFF-SPs does exceed the SSP3-7.0 radiative forcing by 2198.

Table 2 shows the percentage of the runs from this analysis that exceed each forcing threshold in 2100 and then again in 2150 and 2300. This shows that the probability of the RFF-SP/FaIR combination

exceeding the 7 and 8.5 W m$^{-2}$ thresholds are significantly higher than the probability of the RFF-SP emissions exceeding the emissions from the respective SSP. The median forcing in 2100 from the RFF-SP/FaIR analysis is 5.1 W m$^{-2}$ (5th to 95th percentiles from 3.3 to 7.1 W m$^{-2}$).

**Table 1 | Probabilities of exceedance of emissions in 2100 for RFF-SP scenarios relative to SSP-RCP scenarios**

| Comparison scenario | Emissions (2100 or cumulative for 2020–2100) | CO$_2$-only | GWP-weighted | Cumulative CO$_2$ by 2100 |
|---|---|---|---|---|
| SSP5-8.5 | 126 Gt CO$_2$ | 0.030% | 0.020% | 0.020% |
|  | 142 Gt CO$_2$e |  |  |  |
|  | 2103 Gt C |  |  |  |
| SSP3-7.0 | 82.7 Gt CO$_2$ | 1.2% | 0.46% | 1.5% |
|  | 108 Gt CO$_2$e |  |  |  |
|  | 1438 Gt C |  |  |  |
| SSP4-6.0 | 21.9 Gt CO$_2$ | 42% | 36% | 26% |
|  | 39.0 Gt CO$_2$e |  |  |  |
|  | 903 Gt C |  |  |  |
| SSP2-4.5 | 9.68 Gt CO$_2$ | 66% | 72% | 41% |
|  | 19.7 Gt CO$_2$e |  |  |  |
|  | 750 Gt C |  |  |  |
| SSP4-3.4 | –14.8 Gt CO$_2$ | 99% | 98% | 88% |
|  | –0.921 Gt CO$_2$e |  |  |  |
|  | 321.7 Gt C |  |  |  |
| SSP1-2.6 | –8.62 Gt CO$_2$ | 97% | 98% | 92% |
|  | –3.07 Gt CO$_2$e |  |  |  |
|  | 269.4 Gt C |  |  |  |
| SSP1-1.9 | –13.9 Gt CO$_2$ | 98% | 99% | 99% |
|  | –8.61 Gt CO$_2$e |  |  |  |
|  | 69.4 Gt C |  |  |  |

The probability that emissions in 2100 (CO$_2$-only or GWP-weighted) or cumulative CO$_2$ between 2020 and 2100) in the 10,000 Resources for the future socioeconomic projections (RFF-SP) scenarios exceed those in six of the shared socioeconomic pathway-representation concentration pathway (SSP-RCP) scenarios. Global warming potentials (GWPs) are from IPCC AR4. The emissions column contains scenario parameters for Gigatons (Gt) CO$_2$, C, and CO$_2$ equivalents (CO$_2$e). Source data and code for reproducing this table are available (see Data availability section for access information).

## Contributions to uncertainty

While this paper is mainly focused on radiative forcing as the appropriate variable to address for concentration scenarios used as inputs to GCMs, temperatures are a more relevant variable for understanding societal impacts. Figure 3 shows the total uncertainty in radiative forcing (Fig. 3a, b) and Fig. 4 shows the total uncertainty in temperature (Fig. 4a, b), and attributes that uncertainty to parametric socioeconomic uncertainty (characterized as emissions uncertainty), parametric climate response uncertainty, and internal climate variability. The median temperature in 2100 from the RFF-SP/FaIR ensemble is 2.8 °C (5th to 95th percentiles from 1.7 to 4.2 °C), with an 88% chance of exceeding 2 °C. The median arrival year for 2 °C is 2050 (5th percentile of 2037, and the 95th percentile does not exceed 2 °C during the time period of the analysis).

## Discussion

Low-probability scenarios have utility for several applications. The relevance of a particular future scenario depends on the application, purpose, and audience of a particular analysis or assessment. However, the relevance also depends on the probability of exceeding the metric of interest, whether it be a given quantity of emissions, concentration of CO$_2$ or CO$_2$-equivalent, total radiative forcing, global temperature, or some other metric. Despite the low probability of exceeding 8.5 W m$^{-2}$

**Table 2 | Probabilities of exceedance of radiative forcing thresholds for RFF-SP scenarios**

| Radiative forcing threshold (W m$^{-2}$) | Year: 2100 | Year: 2150 | Year: 2300 |
|---|---|---|---|
| 8.5 | 0.53% | 6.9% | 20% |
| 7.0 | 7.2% | 22% | 32% |
| 6.0 | 25% | 37% | 43% |
| 4.5 | 68% | 65% | 62% |
| 3.4 | 92% | 85% | 77% |
| 2.6 | 99% | 95% | 86% |
| 1.9 | 100% | 98% | 91% |

Percent of FaIR/resources for the future socioeconomic projections (RFF-SP) scenarios that exceed each shared socioeconomic pathway-representation concentration pathway (SSP-RCP) radiative forcing threshold in 2100, 2150, or 2300. Source data and code for reproducing this table are available (see Data availability section for access information).

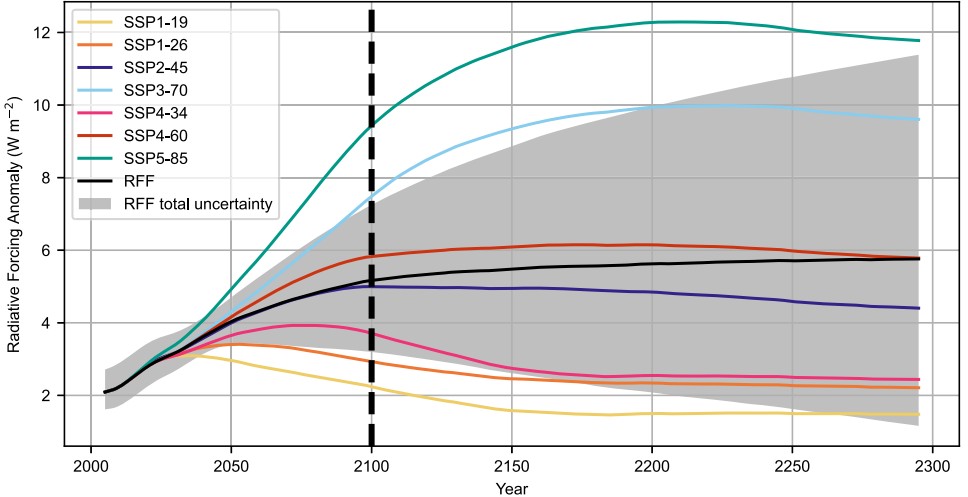

**Fig. 2 | Radiative forcing from RFF-SPs and SSPs.** Mean and 5th to 95th percentiles for total radiative forcing from the Resources for the future socioeconomic projections (RFF-SP) scenarios run through FaIR 2.1 with the 1001 parameter sets, as well as total radiative forcing from six shared socioeconomic pathway-representation concentration pathway (SSP-RCPs), emission scenarios run through FaIR 2.1. Source data are provided as a Source Data file.

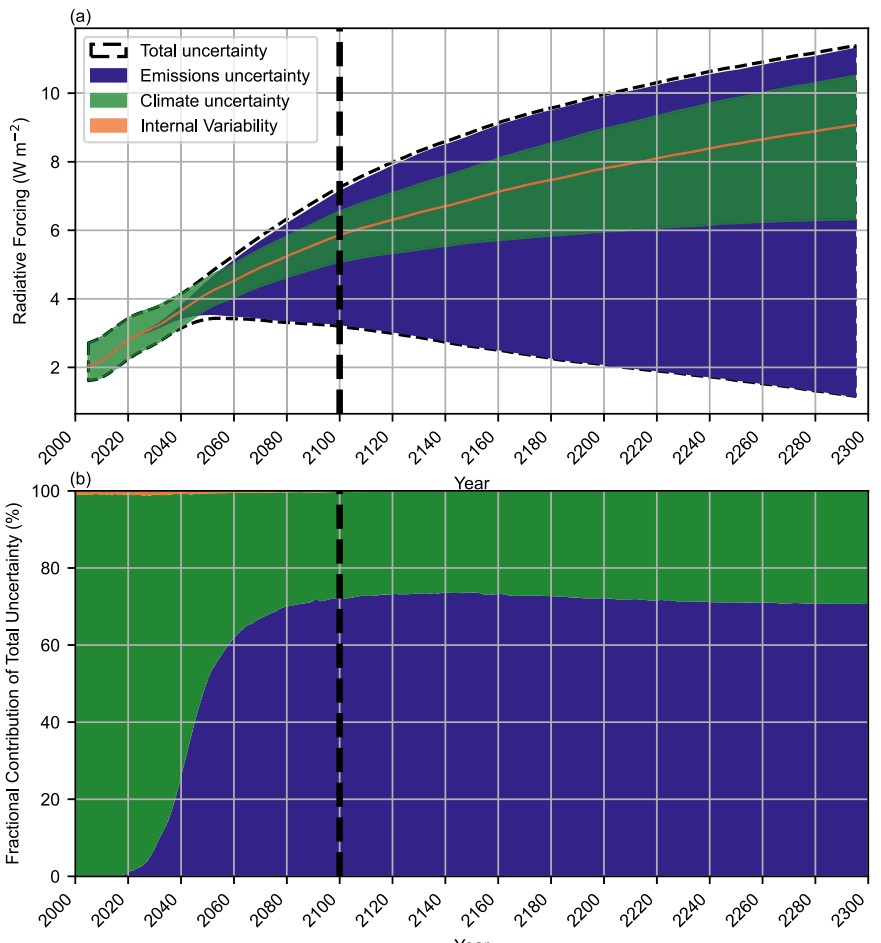

**Fig. 3 | RFF-SP radiative forcing uncertainty.** Uncertainty in effective radiative forcing projections, showing the 5th to 95th percentiles from emissions uncertainty, climate parameter uncertainty, internal climate variability, and total uncertainty in (**a**), and the relative contribution to total uncertainty in (**b**). Source data are provided as a Source Data file.

by the end of the century (Table 2), there are at least three applications where it remains crucial to consider these high-forcing, low-probability scenarios: (1) development of damage functions or "by-degree" climate impact analyses, (2) as an analog for post-2100 climate change, and (3) consideration of low probability but high impact possibilities.

For the first application, "by-degree" climate impact analyses are mostly scenario agnostic[23]. However, the higher the emissions driving the analysis (i.e., a high radiative forcing 8.5 W m$^{-2}$ scenario), the larger the temperature (or other climate parameter) range that will be covered, which is desirable for the purpose of creating damage functions. As damage functions are essentially a mathematical fit to a "by-degree" approach, the anchor for the high end of the damage function should be a low-probability scenario. For example, ref. 24 uses piecewise linear extrapolation between data points to assess climate change damages across a given range of temperature scenarios. Tebaldi et al.[25] note that for a time-shift pattern-scaling approach similar to the Sarofim et al. methodology[23], it is impossible to extrapolate beyond the highest scenario. While the linear fit pattern-scaling methodology can extrapolate beyond the highest scenario directly modeled, Wells et al.[26] find that the use of a warmer scenario to calibrate the linear fit is more useful because of the larger signal-to-noise ratio. Similarly, ref. 27 call for including a high-emission "framing" climate scenario that could represent "the emission world avoided" (or TEWA), in order to aid in understanding how the climate system might change in a high forcing or a high warming world.

The second application for post-2100 analogs is relevant because the majority of GCM scenario projections end in 2100[4]. Therefore, the majority of the impacts literature is also based on scenarios that end in 2100. However, the real world will continue past 2100. For this reason, we argue that climate projections for 2100 under high-forcing scenarios can be used as analogs for climates that may not be realized until much later. This use of climate projections for one year as an analog for climate impacts in a different year is the fundamental concept behind the "temperature slice" or "global warming level" approach[23,25,28]. Given this application, we have calculated the probability density function of radiative forcing in 2150 to show that the probability of exceeding 8.5 W m$^{-2}$ reaches almost 7% due to a combination of potential continued emissions and carbon cycle and methane responses. While the 0.5% probability of exceeding 8.5 W m$^{-2}$ is below the 1% relevance threshold we discuss below, it is difficult to argue policy irrelevance with a probability of exceedance of greater than 5%, even if the date is further in the future. While we have chosen 2150 to illustrate this point, we will note that projections developed for the social cost of greenhouse gas extend through 2300[19], and the exceedance probability of 8.5 W m$^{-2}$ is 20% by the end of this period.

For the third application, low-probability but high-impact events are often of relevance to policymakers. As discussed further in the following paragraphs, the precise probability of exceedance becomes relevant in assessing risk in these contexts. This discussion of probability is also relevant to the choice of the high-end anchor for damage functions or by-degree analyses and for deciding the hottest post-2100 potential climate for which to simulate an analog. The question then becomes at what likelihood does a scenario become relevant. This depends in large part on the impact of the event in question. On the

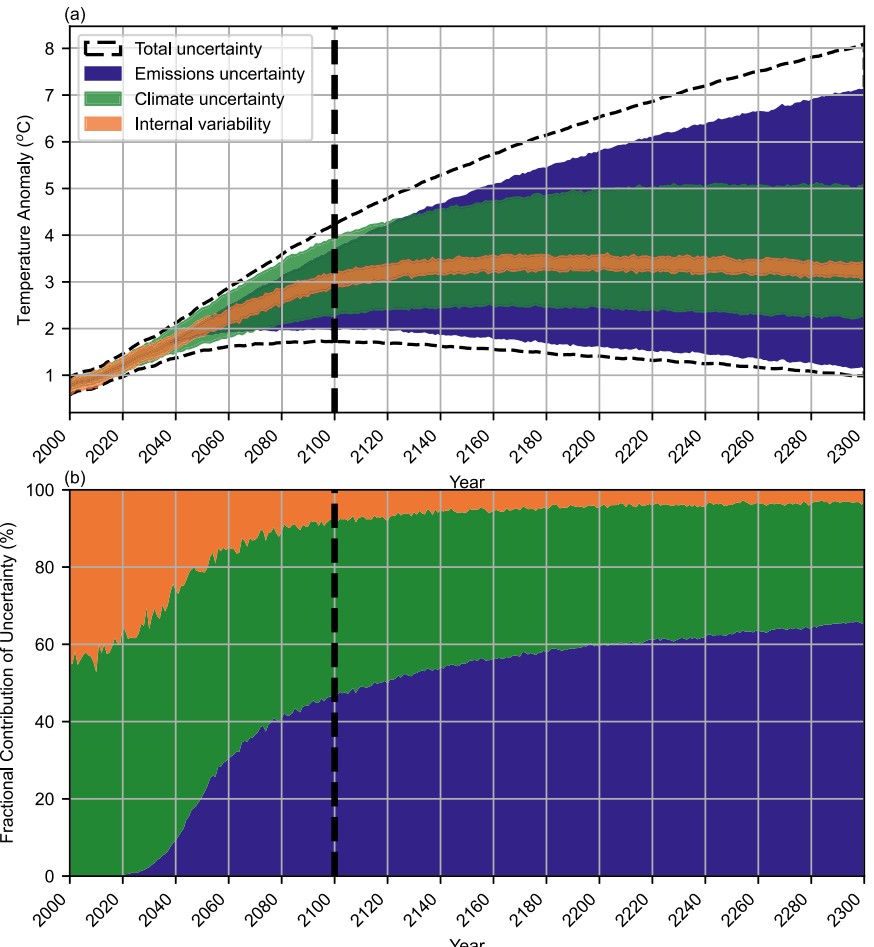

**Fig. 4 | RFF-SP temperature uncertainty.** Uncertainty in temperature projections, showing the 5th to 95th percentiles from emissions uncertainty, climate parameter uncertainty, internal climate variability, and total uncertainty in (**a**), and the relative contribution to total uncertainty in (**b**). Source data are provided as a Source Data file.

conservative end, Ansolabehere et al.[29] recommended an annual risk of 1 in a million for a large, early release of radioactivity from a nuclear plant. On the risk-accepting end, few people would carry an umbrella with them if the probability of rain was less than 10%. However, these examples do not provide a good context for the appropriate choice of likelihood for a high-end climate scenario. Many potential examples involve repeated probabilities for individual events that can happen at multiple locations (e.g., the nuclear example or the choice of the 1 in 100-year flooding event for floodplain analysis). In contrast, climate change involves a global, accumulating problem where the probability of a high-end scenario is a function of socioeconomic assumptions and uncertainty in basic scientific parameters. We choose 1% as a policy relevant probability for a high-end scenario for generalized climate analysis purposes, though recognizing that others could choose a different threshold. Several considerations are relevant for these three applications. There is the choice to focus on the probability of exceeding radiative forcing rather than emissions. There are challenges involved in characterizing uncertainty, particularly for low-probability tails of distributions. And there is the recognition that even if there are good reasons to include a high scenario in an analysis, that scenario is not necessarily the most plausible scenario.

Because CMIP GCMs and ESMs have historically generally been driven by concentration scenarios, not emission scenarios, it is the probability of exceeding those concentrations which is most relevant. Radiative forcing is the best available metric to use to represent the aggregate climate impact of a concentration pathway, though it has

some drawbacks. For example, different GCMs can have different radiative physics, such that different GCMs run with the same set of GHG and aerosol concentrations may result in different levels of forcing, and two emission scenarios with the same forcing in one GCM may diverge when run in a second. While the IIASA database for scenarios suggests that RCP8.5 reaches 8.4 W m$^{-2}$ in 2100 and SSP5-8.5 reaches 8.7 W m$^{-2}$, more recent simple climate models incorporating updated knowledge in Earth system processes yield higher radiative forcings from the same set of concentrations[30]. The most relevant force is the one embodied within the GCM calculations. Fredriksen et al.[31] show that GCM-derived forcing is markedly higher for SSP5-8.5 in CMIP6 models than for RCP8.5 in CMIP5 models. Tebaldi et al.[32] estimate that most GCMs reach 8.9 W m$^{-2}$ by 2100 for SSP5-8.5. Wyser et al.[33] estimated 9.2 W m$^{-2}$ in the EC-Earth3-Veg GCM for SSP5-8.5, but only 7.8 W m$^{-2}$ for RCP8.5. Despite its imperfections, here we focus on radiative forcing as the best available metric.

When looking at low-probability outcomes, considerations of additional sources of unbiased (no more likely to be above the reference parameters than below, and uncorrelated with other parameters) uncertainty will always increase the probability of exceeding those outcomes. For example, while the chance of the world having higher forcing than the SSP5-8.5 scenario in 2100 is a low probability outcome, it is more likely than the chance of exceeding SSP5-8.5 emissions. In this case, the probability of exceeding the emissions in the SSP5-8.5 scenario in 2100 when taking the RFF-SP distribution as truth is 3 in 10,000. Even the probability of exceeding SSP3-7.0 emissions by

2100 is on the order of 1%. However, when we move to radiative forcing, the probabilities become substantially larger: 0.5% for exceeding 8.5 W m$^{-2}$, and 7% for exceeding 7 W m$^{-2}$. A key contributor to this increase in the probability of exceedance is the inclusion of additional uncertainties, in particular, carbon cycle and methane feedback. As a first-order effect, adding an additional uncertainty to an existing probability distribution will lead to an increase in the size of the tails of the distribution.

Over a timescale of several decades, climate parameter uncertainty is the dominant driver of uncertainty in radiative forcing as emissions pathways have yet to diverge substantially due to economic inertia, whereas the rate of carbon uptake and methane atmospheric chemistry both contribute to uncertainty in future concentrations (Fig. 3a, b). However, by mid-century, emissions uncertainty becomes the dominant driver of radiative uncertainty as by 2050, the 95th percentile of $CO_2$ emissions is six times the 5th percentile. In contrast, for temperature, internal variability is initially a substantial component of temperature uncertainty, becoming a smaller proportion of the total over time despite remaining relatively constant in absolute terms, as in ref. 34. Year-to-year variability in temperature due to factors such as ocean currents are of the same magnitude (about 0.2 °C) as the recent decadal trend in temperature. Meanwhile, climate uncertainty continues to be a larger contributor than emission uncertainty through the end of the century (Fig. 4a, b). This is a result of fewer uncertain climate parameters that impact radiative forcing (e.g., aerosol forcing, carbon cycle, and methane feedback uncertainties) than impact temperature (e.g., all the parameters that impact radiative forcing also matter for temperature, but uncertainties in climate sensitivity and ocean heat uptake also have substantial temperature impacts). Due to the cumulative nature of carbon in the atmosphere, the contribution of emissions uncertainty continues to grow with time, becoming the most important factor for understanding temperature trends on a multi-century scale. While most of this paper discusses radiative forcing as it is the key input to climate models, temperature uncertainty is important as temperature drives impacts that are relevant to human and ecological experiences. Two key advances over the previous refs. 34,35 studies use fully probabilistic emission and climate parameter uncertainty inputs for this analysis (rather than, for example, assuming that the RCPs are all equally likely, or that all climate models are equally likely, as was done for the previous analyses).

One noteworthy finding is that the estimated likelihood of high-forcing scenarios has declined over time. While there was no formal probability assigned to the 8.5 W m$^{-2}$ scenario, it was chosen to represent the upper 10% of existing scenarios[1]. Sokolov et al.[36] combined probabilistic estimates of emissions and climate parameters to estimate a 95th percentile of 9.8 W m$^{-2}$ in 2100 relative to 1990 (or about 11.6 W m$^{-2}$ relative to preindustrial). Therefore, over the past 14 years, the estimated probability of exceeding 8.5 W m$^{-2}$ by the end of the century has dropped from double digits to below one percent. Meanwhile, the median temperature projection which was estimated to be as high as 5.1 °C 15 years ago[36], is estimated to be only 2.9 °C in this analysis, with a probability of staying below 2 °C of 12%. While some of this reduction of the high end of possible forcing and temperature futures is due to better information about climate uncertainty, the majority is due to reductions in estimated future emissions, due to a combination of technological progress and policy action. This is good news, and the trend toward a lower probability of high-forcing scenarios will hopefully continue in the future.

Projecting conditions in 2100, fewer than 80 years into the future, is challenging, particularly for socioeconomic variables that are key factors in determining future emissions. Barron[37] discusses some of the challenges involved in projecting future penetration of low-carbon technologies using climate policy models. Projecting socioeconomic conditions and emission drivers through 2300 is an even more challenging task, and yet is necessary for considering the long-term

implications of long-lived GHG emissions. The Meinshausen et al.[38] approach to extending the SSPs uses stylized assumptions about emissions converging to zero by 2250. While the RFF-SP scenarios may be the best probabilistic emissions forecast available, these probabilities must be taken with some skepticism, particularly in the tails (such as the 1% threshold that we focus on). The RFF-SPs rely primarily on expert elicitation, but there are other approaches. Liu and Raftery[39] use a Bayesian approach to drive a Markov chain Monte Carlo for developing emission scenarios through 2100. Liu and Raftery's distribution in 2100 is very similar to that of the RFF-SPs for the 95th percentile, but the median of their distribution is almost double that of the RFF-SPs, and the 5th percentile is greater than 10 GtCO2/year, whereas the RFF-SPs have a 5th percentile that is negative. Morris et al.[40] apply a complex computable general equilibrium model driven with uncertain input parameters whose distributions are defined by a combination of expert elicitation and statistical analysis of historical data, also through the year 2100. However, Morris et al.[40] separate policy uncertainty out as a separate factor with four possible futures, making it difficult to compare directly to the RFF-SP approach, which treats policy probabilistically. Addressing the potential for future policy in a probabilistic framework is challenging—for the RFF-SP scenarios, the experts were instructed to include views about the evolution of future policy for projecting changes in technology, fuel use, and other socioeconomic conditions relevant to emission factors.

While there is more confidence in the climate uncertainty probabilities as expressed in the 1001 FaIR ensemble members compared with the socioeconomic uncertainties driving emission projections, there are also some limitations regarding these physical parameter definitions. In particular, any feedback that doesn't exist in either the historical record or the climate models against which FaIR was calibrated will not be captured by this analysis. For example, the permafrost feedback is not included in FaIR, which has the potential to contribute substantially to radiative forcing and warming through the release of additional $CO_2$ and $CH_4$. Furthermore, prior distributions for FaIR parameters were informed by CMIP6 GCMs, and the potential for the uncertainty space to be under-sampled exists.

While our analysis shows that 8.5 W m$^{-2}$ scenarios are appropriate for use in creating damage functions, and important to include in the next round of CMIP analyses, it is important to reiterate that 8.5 W m$^{-2}$ is still an unlikely future outcome—even though the 22nd century. We are arguing here for the inclusion of 8.5 W m$^{-2}$ scenarios to anchor the high end of possible outcomes or to develop damage functions, but not as the most likely future. One pertinent example is the use of a future scenario for use in adaptation decisions. There are often non-negligible costs associated with over-adaptation: e.g., for sea level rise, there are expenses involved in building higher sea walls than would be optimal from a cost-benefit analysis, and in some cases, there could be unnecessary abandonment of properties based on over-reliance on pessimistic forecasts. While it may be appropriate to use a scenario that is higher than the median or mean scenario, the 25% likely 6 W m$^{-2}$ scenario might be better suited than the 0.5% likely 8.5 W m$^{-2}$ scenario (Table 2). However, this does not mean an 8.5 W m$^{-2}$ scenario cannot be relevant: the use of a by-degree approach allows for disassociating the climate model output from the climate model time period. This approach—referred to as "time-slices"[28], "time-shift"[41], or "temperature binning"[23]—allows for the GCM output of early decades from a high emissions scenario to be used as analogs for temperatures in later decades of lower emissions scenarios. Reduced complexity approaches (such as the combination of FaIR and the RFF-SPs) can be used to determine the arrival times of these temperatures: moreover, these approaches can also produce the probabilities of exceedance of different temperature thresholds, which would be useful information for adaptation decisionmakers and other policymakers.

This kind of probabilistic analysis is made possible by the use of thousands of probabilistic economic scenarios coupled with

thousands of probabilistic parameter sets from a reduced complexity climate model. Given the computational demands of full-complexity GCMs, the scenario community has generally chosen fewer than a dozen primary scenarios to drive these GCMs. A proper probabilistic analysis would be impossible with so few emissions or concentration scenarios, and then the GCMs themselves are not designed for probabilistic analysis[42]. Therefore, the kind of probabilistic analysis highlighted in this paper is designed as a complement to the more traditional scenario-based approach, which produces a finite set of alternative future storylines without assigned probabilities[10].

In summary, here we use a 10 million-member ensemble to analyze the probability of exceeding 8.5 W m$^{-2}$ in 2100, 2150, and 2300. Our results show that while the probability of exceeding this threshold in 2100 is less than 1%, it reaches 7% by 2150, and 20% by 2300. Because many climate analyses to date have focused on the current century, a scenario that reaches 8.5 W m$^{-2}$ in 2100 continues to have substantial relevance for serving as an analog for post-2100 climates, as well as for driving impacts models to develop damage functions that can be useful through the year 2300.

Therefore, it is important that the CMIP community continue to include an 8.5 W m$^{-2}$ scenario in the family of scenarios for CMIP7. In addition, if an impact modeler is only going to use a single scenario, 8.5 W m$^{-2}$ is the appropriate scenario to use—not because it is the most likely, but because it can serve to inform a by-degree approach which can be coupled with a reduced complexity model to estimate the timing and probability of reaching future forcing and temperature thresholds, enabling the discussion of almost any possible future including years past 2100 (with a possible exception of peak-and-decline scenarios). However, if it becomes standard in the future to run impact analyses through later years, such as 2125[2], it might be more appropriate to choose a high-end scenario which reaches 8.5 W m$^{-2}$ in that later year, rather than 2100.

While we argue that 8.5 W m$^{-2}$ continues to be relevant based on our probabilistic analysis, we can hope that as zero carbon technologies continue to become more competitive, and mitigation efforts continue to increase, the 8.5 W m$^{-2}$ scenario may eventually drop enough in likelihood that it would no longer exceed our 1% threshold in any year and could be discarded.

## Methods
We quantify the likelihood of exceeding 8.5 W m$^{-2}$ by the end of the century by developing a probability density function of radiative forcing, based on two components: (1) probabilistic projections of greenhouse gas (GHG) and aerosol emissions and (2) reduced complexity climate model that accounts for uncertainty in climate response.

### Generation of emission projections
To generate emission projections, we start with the 10,000 equally probable global GHG emission scenarios from the RFF-SP dataset, which provides global $CO_2$, $CH_4$, and $N_2O$ emissions from 2020 through 2300[20]. RFF-SP is recent, open-source, fully probabilistic, extends through the year 2300, and is used by the U.S. SC-GHG process, making it an ideal dataset for emission projections. While the original RFF-SP scenarios assumed that additional short-lived climate forcers (tropospheric ozone and aerosol precursors) and other greenhouse gases followed the SSP2-4.5 scenario, here we infill the RFF-SP GHG scenarios using the silicone tool[21]. Infilling uses a database of existing integrated assessment model (IAM) GHG and air pollutant emissions scenarios and uses relationships between a lead species (in our case, $CO_2$ emissions from fossil fuels and industry) and those that are not present in the RFF-SPs. The logic behind this is that emissions of $CO_2$ tend to be well-correlated with emissions of non-$CO_2$ species over time[43] as many species are co-emitted with $CO_2$. Where they are not, high emissions scenarios tend to be indicative of high economic activity, low levels of climate and air quality policy, or both (and vice

versa for low emissions scenarios). In this analysis, we use the IPCC Sixth Assessment Report (AR6) Working Group 3 (WG3) infiller database[30,44]. The emissions in this database have been harmonized to ensure a smooth transition from historically reported emissions values to future scenario projections[45]. As the RFF-SP dataset only provides total $CO_2$ emissions, this contribution needs to be split into fossil fuel and industrial (FFI) emissions and land use (AFOLU) emissions to run FaIR and perform the infilling. This was performed using the silicone toolbox with the "time-dependent ratio" method, mapping FFI and AFOLU emissions to total $CO_2$ emissions from the AR6 infiller database using the average ratio of AFOLU and FFI to total $CO_2$ across the database[21].

### Infilling gases beyond $CO_2$, $N_2O$, and $CH_4$
We replicate the infilling methods of the IPCC Working Group 3 report[44,46] as closely as possible. For example, short-lived climate forcers (SLCFs: $SO_2$, BC, OC, $NH_3$, $NO_x$, CO, and VOCs) are infilled using the "quantile rolling windows" (QRW) method of silicone. We use $CO_2$ FFI as the lead species and the AR6 infiller database for both $CO_2$ and SLCF emissions. QRW maps the lead species ($CO_2$ FFI) to a specified quantile of emissions of the desired pollutant from the infiller database over time. We use the 50th percentile of emissions across all scenarios in the AR6 infiller database for each short-lived forcer. This preserves relationships of co-emittance between high and low emissions scenarios (e.g., high $CO_2$ FFI likely implies high $SO_2$ emissions in the absence of air quality improvements) but does not further differentiate between scenarios (e.g., high $CO_2$ FFI emissions could occur with low $SO_2$ emissions if air quality improvements occur but emissions mitigation does not).

All halogenated GHG emissions are infilled using the "RMS Closest" method of silicone. As suggested, this selects the emissions of the desired greenhouse gas from the AR6 scenario based on the scenario that minimizes the root-mean-square error between the timeseries of (reconstructed) $CO_2$ FFI emissions from the RFF-SP scenario and the AR6 scenario. For some of the more major hydrofluorocarbons (HFCs) and perfluorocarbons (PFCs) that are individually modeled by several IAM groups, the AR6 infiller database is used. In contrast, emissions for more minor gases and all ozone-depleting substances are not generally available in IAMs, and therefore the set of eight SSP-RCP scenarios that drive the CMIP6 Earth System models were prepared on global annual mean emissions timeseries for the Reduced Model Intercomparison Project (RCMIP)[47,48]. FaIR uses NOx emissions from the aviation sector as a proxy for radiative forcing from contrails and contrail-induced cirrus, and this data is also infilled from the SSP-RCP scenarios using the RMS Closest method.

In addition, the AR6 and SSP-RCP scenarios only run to 2100, whereas we require emissions to 2300. We follow the logic discussed in Meinshausen et al.[38] to extend SSP-RCP emissions beyond 2100. If $CO_2$ FFI emissions are positive in 2100, they are linearly ramped down to zero by 2250; if they are negative, they are ramped up to zero by 2150. $CO_2$ AFOLU emissions are ramped to zero by 2150 whether negative or positive, in 2100. For $CH_4$, $N_2O$, and short-lived forcers, the FFI component (always positive) is ramped down to zero by 2250, and the AFOLU component is held constant at 2100 emissions. As IAM scenario data does not provide the FFI/AFOLU split for non-$CO_2$ emissions, we estimate this from the average of the 2100 ratios in SSP-RCP scenarios that do have this granularity. Finally, minor greenhouse gases with no assumed AFOLU source are ramped down to their preindustrial emissions level by 2250. Historical emissions from 1750–2014 are provided from RCMIP and for the period 2015–2020 we follow the RFF-SP logic and use SSP2-4.5.

### Using the FaIR model for temperature projections
The final set of 10,000 infilled, complete RFF-SP emissions scenarios are then run using a 1001-member probabilistic ensemble of a reduced

complexity climate model: the Finite amplitude Impulse Response model (FaIR v2.1)[22]. The FaIR model is a widely used reduced complexity model that has been relied on by major institutions such as the IPCC[49], the US National Academies of Science[50], and the EPA Social Cost of Carbon[50], and has performed well in the Reduced Complexity Model Intercomparison Project (RCMIP)[47]. The parameters for the model were calibrated to be consistent with historical data taking into account observational uncertainty (ocean heat content, $CO_2$ concentrations, aerosol radiative forcing) as well as consistency with the ranges for transient climate response and equilibrium climate sensitivity presented in the IPCC AR6 Working Group 1 assessment[49,51,52]. Each member of the ensemble maintains a root-mean-square error between the modeled and observed global mean surface temperature of less than 0.16 °C and is consistent with historical emissions from RCMIP. We use stochastic, auto-correlated internal variability in temperature and forcing[53]. For comparison to the RFF-SP dataset, we also run the SSP-RCP scenarios under the same calibration for comparison[54].

### Comparison and assessment

For comparisons of emission levels between RFF-SP and SSP-RCP scenarios (Fig. 1 and Table 1), we calculate the number of RFF-SP scenarios where $CO_2$ emissions in 2100 exceed those in the six SSP-RCP scenarios, and then divide by the 10,000 total scenarios to come up with a probability of exceedance. This calculation is repeated with global warming potential (GWP)-weighted emissions to account for the role of non-$CO_2$ GHGs, using AR4 GWPs. The calculation is repeated a third time for cumulative $CO_2$ emissions between 2020 and the year 2100.

The RFF-SP emissions are then used as inputs to the FaIR model. Each emissions scenario is run with all 1001 FaIR parameter combinations. The coupled combination of 10,000 RFF-SP emissions scenarios, run with a 1001-member climate parameter ensemble, creates 10,010,000 total equally probable outcomes (https://zenodo.org/record/7759089#.ZGI7kqXMK3C). Result data for each emissions scenario are saved as a netCDF file and include ensemble-level information about global mean surface temperature, effective radiative forcing, ocean heat content, and concentrations of $CO_2$, $CH_4$, and $N_2O$.

For comparisons of radiative forcing between the RFF-SP and SSP-RCP scenarios, we use this full 10,010,000 ensemble. For Fig. 2, we compare the SSP-RCP scenarios (run through FaIR with the same calibration) to the RFF-SP/FaIR analysis. For Table 2, we report the percentage of RFF-SP/FaIR scenarios that exceed each given forcing threshold in 2100, 2150, or 2300.

To assess the relative contributions of emissions uncertainty and climate parameter uncertainty, we also produce Figs. 3, 4, in the style of refs. 34,35. To assess the contribution of internal variability, we also produce parallel FaIR ensembles both with internal variability switched on (stochastic) and switched off (deterministic) Figs. 3, 4 show a timeseries of the range of radiative forcing and temperatures (an output from FaIR) resulting from the 1001 FaIR runs around the emissions scenario with median temperature (climate uncertainty), the median temperature projection from each of the 10,000 RFF-SP runs (approximately corresponding to parameter sets with the median climate sensitivity; emissions uncertainty), the contribution of internal variability defined as the uncertainty across the same median RFF-SP scenario used to determine climate uncertainty, subtracting the stochastic ensemble from the deterministic ensemble, and the total uncertainty, along with the fractional contribution of each. In all cases, we take the 5th to 95th percentiles of the resulting distributions to report the uncertainties. As a completely clean separation of components is difficult and there is likely some overlap, the sum of the individual uncertainty components in quadrature is expected to exceed the total uncertainty.

## Data availability

RFF-SP emissions scenarios are available from https://doi.org/10.5281/zenodo.5898729. The AR6 Working Group 3 infilling database is available from https://doi.org/10.5281/zenodo.6390768 (registration required). RCMIP emissions prepared for the SSP-RCP scenarios are available from https://doi.org/10.5281/zenodo.4589756. The probabilistic FaIR output is available from https://zenodo.org/record/7838148. Source data are provided with this paper.

## Code availability

FaIR v2.1 is available from https://doi.org/10.5281/zenodo.7459702 and v1.0 of the calibration is available from (https://doi.org/10.5281/zenodo.7545157). Code for processing probabilistic FaIR output is available at https://github.com/chrisroadmap/rff-fair2.1. The code for FaIR/Figure processing scripts is at https://github.com/erysimumcap/RCM_RCP8.5_Probability.

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

## Acknowledgements

This research was funded by the US Environmental Protection Agency under contract #68HERH19D0027. The views expressed in this article are those of the authors and do not necessarily represent the views or policies of the US Environmental Protection Agency.

## Author contributions

M.C.S. conceived of the project and designed the study. C.J.S. performed the FaIR climate calculations. P.M. did the data analysis and generated the figures. C.R.L. managed the project. M.C.S., C.J.S., P.M.,

S.M., C.A.H., and E.E.M. contributed to discussions about the research, as well as helped with writing and editing the manuscript.

## Competing interests

The authors declare no competing interests.
