## [Peer Review File · Nature Communications]

REVIEWER COMMENTS

Reviewer #1 (Remarks to the Author):

General comments:

This is an interesting paper which combines a published ensemble of socioeconomic scenario with an ensemble of climate models. The authors assess what fraction of the ~10-million resulting scenarios exceed 8.5W/m^2 at different times, looking to 2300. They find that 8.5W/m^2 is unlikely: very unlikely by 2100 and moderately unlikely by 2300. They discuss some reasons why researchers might use RCP8.5 or SSP5-8.5 anyway, and call for those scenarios to be used as high-end anchor scenarios, but not as probable or plausible ones. The paper contributes to growing and important literatures on high-emission scenarios and plausible climate futures.

I think the paper has some interesting and valuable modeling results, but it needs a revision to bring these results more to the surface. I am not sure that these results quite rise to the level of Nature Communications, but they deserve to be published in a top field journal such as Communications Earth & Environment or Environmental Research Letters. Below, I offer some comments that I hope will be helpful in a revision. I congratulate the authors on a nice analysis, which makes a valuable contribution to the literature.

1. The Results section currently reads like a methods section, and the Discussion section mixes results and discussion. The figures provide a helpful summary of the modeling results but the text does not summarize them. The discussion of when to use RCP8.5 or SSP5-8.5, despite their low probability, is interesting and sensible, but is not as novel nor as important as the modeling results are, in my view. Therefore, I suggest focusing the discussion more on the modeling results.

2. The text should make it clearer what the substantive differences in mechanistic insight and results are of the present analysis, compared to Rennert et al. Nature (cited), which also conducts a probabilistic analysis of future CO₂ emissions and temperature. The present analysis includes minor GHGs and aerosols. Does its climate model also assess different sources of climate sensitivity? It seems like the modal forcing in the present study is slightly higher than in Rennert et al. Is that difference due to accounting for aerosols? If so, this would be very interesting in light of the recent public debate (featuring Hansen, Mann, and others) regarding the role of aerosol pollution reduction in the future warming outlook. Please elaborate on these differences and their causes, as they are potentially an important contribution of the paper.

3. The paper should discuss its ‘probabilistic’ projections a bit more circumspectly. The Rennert/Raftery/Müller et al./RFF methods for probabilistically projecting emissions and their drivers are the best we currently have, but assessing probabilities for these quantities out to 2100, and even more so to 2300, is still extremely fraught and subject to deep and unquantifiable uncertainties. I don’t think it is defensible to take such probabilistic projections literally, even though the density of scenarios does provide some valuable information that is partly related to probability.

4. Relatedly, the extensions of socioeconomic scenarios from 2100 to 2300 used in this study are extremely speculative. This should also be acknowledged.

5. Lastly, to the extent space allows, it would be helpful to discuss the mechanistic aspects of the RCP8.5/SSP5-8.5 debate a bit, in context with the model’s results. Probabilistic projections based on historical trends, such as from RFF, etc., are valuable, but should also be discussed in context with mechanistic questions about the plausibility of RCP8.5/SSP5-8.5’s coal assumptions (see Ritchie and Dowlatabadi 2017 *Energy* 140, 1276, Pielke and Ritchie 2021 *Energy Research & Social Science*, 72, 101890, and Burgess et al. ERL 2020 already cited), SSP5’s economic assumptions (see Burgess et al. 2023 *Communications Earth & Environment* 4, 220), SSP5-8.5’s land-use assumptions (see Schwalm et al. 2020, already cited, and Hausfather and Peters 2020 *PNAS* 117, 27791), etc.

Specific comments:

Line 53: “RCP8.5 has been characterized as a scenario that assumes high population growth, low income growth, and modest improvements in energy intensity over the next century”. The authors should note that these population growth and income growth assumptions are reversed in SSP5-8.5 (very high income growth and low population growth), and the high income growth assumption is key to the energy demand in SSP5-8.5; see Riahi et al. (2017).

Line 98: Define ‘GWP’ (global warming potential, I think, but the same acronym is also commonly used for gross world product) at first mention.

Line 116: There are a few other rationales for using (or for not using) high-emission scenarios that have been proposed. Some are reviewed by Burgess et al. (2023) *ICES Journal of Marine Science* 80, 1163.

Line 142: Deep uncertainty and discounting also affect the policy relevance of warming that may occur in 2150 and beyond. Analogously looking back to 1890, it is difficult to imagine how many of today's policy challenges would have been predictable to the people then—including the climate challenges and technologies available for adaptation and mitigation. The same goes—even more so—for 2300.

Line 146: There is a rich literature in economics on how decision-makers should handle low-probability, high-impact events. I suggest looking at and discussing Weitzman (2014) *American Economic Review* 104, 544, Costello et al. (2010), *PNAS* 107, 8108, and Anthoff and Tol (2022) *Journal of the Association of Environmental and Resource Economists*, 9, 885.

Line 194: To what extent does the paper's ensemble capture uncertainty in climate-system feedbacks? This is a possibility often discussed by proponents of using RCP8.5 as a prominent scenario.

Line 266: This isn't necessarily true if the timing of the warming matters to the impact.

Line 274: It is not clear what on what basis this claim is made or on what basis future evidence could reverse it. What probability threshold is being assumed?

Reviewer #3 (Remarks to the Author):

A review of the manuscript entitled "How high is high enough? A multi-million-member ensemble analysis of future climate scenarios and their relevance" by Sarofim et al.

The authors used a 10 million-member-ensemble plausible future realizations to examine the possibility of exceeding 8.5W/m² in a future warming climate. They found that the probability of exceeding this threshold in 2100 is less than 1%. Though with an extremely low probability, the authors argued that it is still of societal relevance for the community to continue to use a scenario that reaches 8.5W/m² in 2100 for serving as an analogue for post-2100 climates and developing damage functions that can be useful through 2300. The results, however, are not surprising and do not represent conceptual novelty over previous studies. I have several concerns about the

interpretation and presentation of the results, which preclude me to recommend acceptance of the paper in its current form for Nature Communications.

Major comments:

(1) Lack of novelty. The main idea in this study that it is extremely unlikely to exceed 8.5W/m^2 by 2100 (i.e., equivalent to RCP8.5 and SSP5-8.5 scenarios) is essentially the same as Hausfather & Peters (2020), except that the present study uses a super large ensemble with multiple members and includes some analyses of uncertainties. However, the uncertainty part (Fig.3) is also a kind of repetitive analysis of Hawkins and Sutton (2009) and Lehner et al. (2020), both of which already concluded that the uncertainty due to internal variability (similar to the climate parameter uncertainty used in this study; Fig.3) is a main driver of near-term climate.

(2) Insufficient explanation of physical mechanisms. The authors spent a large part of text on “Discussion”, which, however, only focused on “implication” rather than the “cause” of the results. For example, while the climate parameter uncertainty and emissions uncertainty equally contribute to total forcing uncertainty by 2050, climate parameter uncertainty exceeds emissions uncertainty almost through 2100 for temperature uncertainty. This is an interesting result of great importance but the authors failed to explain why, for instance, by which physical processes and/or through which climate variability modes this would occur. The provided explanation (Line198) is rather mathematical than physical.

(3) Ambiguity of methods. The definition of the three types of uncertainty is difficult to understand. For example, what do climate parameters specifically represent? Internal variability? Or model physics that determine certain oceanic and atmospheric feedbacks? Or others? Which climate parameter is used in calculating emissions uncertainty? Which single emissions scenario is used to compute climate uncertainty? There lacks clear step-by-step description of procedures and associated rationales for calculating the uncertainties.

(4) Related to my concern above, to what extent does internal variability contribute to each of the three uncertainties? what is the role of internal variability in partitioning/affecting the uncertainty contribution? Why the uncertainty range (gray shading) initially increases, then peaks around 2100 and slightly decreases thereafter in CO₂ emissions (Fig.2), while the uncertainty range continuously increase through 2300 for radiative forcing (Fig.3)? In Fig.3a, after ~2080, emissions uncertainty nearly accounts for 100% of total combined uncertainty, while climate parameter uncertainty remains large (>40%), how to interpret this? In Fig.3b, why in ~2030 the fractional contribution of climate uncertainty can even exceed 100%?

(5) The authors noted that internal variability contributes to all the three uncertainty distributions. However, the internal variability may be vastly different in each uncertainty, making it impossible to compare the two emissions and climate uncertainties. For example, at a certain year when emissions uncertainty exceeds climate uncertainty, internal variability may play a bigger role in the emissions uncertainty. As such, when the influence from internal variability is excluded for both cases or exactly the same internal variability is used for each case, the “pure” emissions uncertainty may not exceed climate uncertainty, which undoubtedly undermines the robustness of the results and conclusions presented in this study.

(6) Line210: as noted by the authors, there are many other different approaches in developing emission scenarios. Why the authors choose to use RFF-SP and consider the RFF-SP to be the best? What is the biggest advantage of RFF-SP?

(7) For the second application, the authors argued that climate projections for 2100 under high forcing scenarios can be used as analogues for climates that may not be seen until much later (Line 135). However, climate impacts are not linearly related to the intensity of forcing scenarios and there are hysteresis-like behaviors to CO₂ forcing. That said, climate projections for 2100 cannot be used as analogues for climate in 2200, even under the same high forcing scenarios. In this case, the associated impacts and mitigation and adaptation strategies are also completely different.

Specific comments:

(1) Line61: what is BAU?

(2) Line98: what is GWP-weighted?

(3) Line104: should be Table2?

(4) All the probability presented in Table1 and 2 should have corresponding uncertainty ranges.

(5) Line150: delete “to be”.

(6) Line159: what is the rationale for choosing 1%? It seems a very arbitrary choice.

(7) Table1: what does “GtCO₂e” represent?

(8) Fig.2: add units in y-axis.

(9) Fig.3: add labels “a, b, c, d” in each panel.

Reviewer #4 (Remarks to the Author):

The authors aim to assess The likelihood and relevance of using RCP8.5, which may be relevant for the scientific community using scenarios for climate and adaptation analysis.

The Abstract and Introduction appear unstructured and confusing. However, with a few changes, you could appear engaging and provide an overview of the relevant context and knowledge gaps and what is intended by the authors (which is now more or less missing).

The paper needs more focus and a major rewrite of the abstract and introduction), and I think the authors write to a very narrow audience (both writing style/explanation level and focus). You could expand and include all your results rather than only focus on 8.5.

I got confused about which scenario series you use (be clear from the beginning) and only realized it in the results/Table 1), so that reflects my comments up till then. I probably made several repetitions in the comments ;)

Abstract:

You could improve the abstract by first presenting the context you are writing into (which area of interest) and the knowledge gap (why is this study important), maybe the method/approach, results, and finally, what the results contribute to in the abstract.

Right now, it is a bit messy ;)

Decide if you wanna use the term emission or forcing (make it simple to read)

You write "calibrated to match the IPCC 6th Assessment Report." What does this mean?

I suggest you add more context, e.g., state why you are doing the study, why it is important, the knowledge gap, and finally, what the results contribute to in this larger context.

Introduction:

The same comment goes for the introduction. Right now, it is a bit messy, and you can improve by introducing the greater context and then going more into detail. And engaging the reader (e.g., introducing RCPs 8.5 as "the highest of four concentration scenarios" could come in sentences 2,3,4 or 5)

I prefer more context to inform (and engage) the reader. I am not sure, but your study objects are the RCPs 4.5 and 8.5. Why are the RCPs interesting/important other than being used for AR5? They were published more than ten years ago. They may still be important for analyzing future CC, and thus, high-impact (low probability) scenarios are important for adaptation analyses.

I think the context (as you mention later) could be the discussions about the relevance of RCP8.5, which is the discussion you seek to participate in - the relevance of your paper's theme. The key or initial paper would be Hausfather/Peters (2020): <https://www.nature.com/articles/d41586-020-00177-3>. Check also Pedersen et al. (2020): <https://www.nature.com/articles/d41586-020-00177-3>.

Reading the Intro, I'm confused about your focus. Are you focusing on RCPs (4.5 and 8.5) or SSP-RCPs? Your research would appear more up-to-date if you assess the SSP5-85 (less likely) instead and compare it with the SSP2-4.5 as "the current policy" case. The SSP3 i is also a high-emission scenario, and the SSP3 narrative is quite similar to recent trends with national conservative moments, increased national security, and national defense budgets.

l. 416: Seeing Table 1, I realize which scenarios you are examining. Make it clear what you are focusing on earlier :)

Why not assess the entire suite from 1.9 to 8.5 if you focus on forcings? It would be highly interesting to know but maybe a lot of work! At least the upper end, intermediate (4.5), and lower ends (1.9 and 2.6) would be interesting also from a policymaker perspective.

I'm still not convinced it is relevant with one more paper assessing 8.5, but maybe I'm wrong. I suggest that you focus on the entire suite and frame it more broadly: can we reach the Paris Agreement, or where are we heading, and can we go as high as 8.5? (this is just a suggestion to make it more policy-relevant and relevant for a larger community. A 70% chance of exceeding 4.5 and 90% exceeding 3.4 is quite interesting, and stating/confirming (SSP2-)4.5 as a "current policy" case. I would change your scope/context :)

In addition, many have assessed the likelihood of these scenarios and the earlier generation. You would show a convincing overview if you justify why you assess "probabilistic emission projections" on scenarios that are not probabilistic or based on likelihoods. Historically the role of scenarios (probabilistic or qualitative scenarios) has been discussed since Schneider (2001) vs. Nakicenovic (2002) - but easiest to mention that there is an overview of this in this 2022-paper: <https://www.sciencedirect.com/science/article/pii/S0959378022000760>.

What does this mean: "given the latest available societal projections."

The method description, 83-92, is quite nice. But be sure to explain what some acronyms stand for and mean as a method, e.g., RFF-SP.

Results:

I am not sure what is RFF-SP scenarios: "We first compare emission levels across the RFF-SP scenarios to those in the SSP scenarios" (l. 95)

This needs to be explained better.

For me, the results are written to an even narrower audience. I would expect at least an explanation of what the results and numbers mean (l 100). It's like you have been writing to your colleagues ;) Practice communicating it to a broader audience and explain what the numbers mean :)

Anyway, explain the results in a few lines. In my opinion, it is not enough to present them in a table (l. 100), although you discuss them later :)

Which “six SSP scenarios” are you analyzing? (l. 97) You would benefit from being more informative about what you are doing and what you are studying and why Introduction to results

In essence, results and discussion works better, are more stringent and less messy, and have better quality than the Abstract and Introduction. Your conclusions about the “probability of exceeding this threshold in 2100 are less than 1%, it reaches 7% by 2150, and 20% by 2300” are interesting, but SSP5-RCP8.5 is mainly interesting related to adaptation assessment as the high-impact (low probability) case. You, of course, add some value here and more numbers, but how does your work differ from, e.g., Christensen's (I think they focused on GDP, right?) - be specific about this in the introductory (which context you are writing into and what your results contribute to). As stated earlier, I would create a new focus using all your results and not focusing only on 8.5.

Detailed comments:

Introduction: create a new beginning where you include the overall context and rationale of the paper :) E.g., don't begin with the RCPs if you are examining the SSP-RCPs

l. 41: US - write the United States for the first time

l. 41-46: Is this an important contribution to the science to mention this as a backup for your choice of scenarios? (I don't know the reference, so I will not define if it is or not) “and the US 42 National Climate Assessment (NCA4) and innumerable research papers. While NCA4 used the full 3 43 range of RCP scenarios, the assessment focused most closely on RCP4.5 and RCP8.5, which they 44 considered as low-end and high-end scenarios, to be consistent with the range of scenarios used in 45 previous assessments (USGCRP, 2015).”

l.83-87 becomes quite technical. I would introduce and explain what RFF-SP is in more detail (1-2 sentences) before using it in an RQ

l.95: “We first compare emission levels across the RFF-SP scenarios to those in the SSP scenarios”

Explain what this means for the less technical reader. Again, at this stage, RFF-SP still appears relatively unknown to the reader, and be more precise: are you assessing the SSP-RCPs via a method, or are the RFF-SP different scenarios (I guess the first, but I'm not sure actually)?

General comments:

This is an interesting paper which combines a published ensemble of socioeconomic scenario with an ensemble of climate models. The authors assess what fraction of the ~10-million resulting scenarios exceed 8.5W/m^2 at different times, looking to 2300. They find that 8.5W/m^2 is unlikely: very unlikely by 2100 and moderately unlikely by 2300. They discuss some reasons why researchers might use RCP8.5 or SSP5-8.5 anyway, and call for those scenarios to be used as high-end anchor scenarios, but not as probable or plausible ones. The paper contributes to growing and important literatures on high-emission scenarios and plausible climate futures.

Thank you for the comments

I think the paper has some interesting and valuable modeling results, but it needs a revision to bring these results more to the surface. I am not sure that these results quite rise to the level of Nature Communications, but they deserve to be published in a top field journal such as Communications Earth & Environment or Environmental Research Letters. Below, I offer some comments that I hope will be helpful in a revision. I congratulate the authors on a nice analysis, which makes a valuable contribution to the literature.

Thank you for the comments: we hope to persuade you that (after revisions) the paper topic is worthy of this journal. In particular, the imminent decisions about scenarios to use as an input to CMIP7 makes this topic particularly timely (e.g., see the ScenarioMIP workshop report from van Vuuren et al 2023).

1. The Results section currently reads like a methods section, and the Discussion section mixes results and discussion. The figures provide a helpful summary of the modeling results but the text does not summarize them. The discussion of when to use RCP8.5 or SSP5-8.5, despite their low probability, is interesting and sensible, but is not as novel nor as important as the modeling results are, in my view. Therefore, I suggest focusing the discussion more on the modeling results.

Thank you. We have revised the text substantially to better differentiate methods, results, and discussion, and added more text discussing the results shown in the figures.

We appreciate the comments regarding the relative importance of the modeling and the SSP5-8.5 usage: again, because of the CMIP7 ScenarioMIP decisions, we feel like the 8.5 discussion is particularly relevant, but we have worked to improve our discussion of the other radiative forcing and temperature thresholds.

2. The text should make it clearer what the substantive differences in mechanistic insight and results are of the present analysis, compared to Rennert et al. Nature (cited), which also conducts a probabilistic analysis of future CO₂ emissions and temperature. The present analysis includes minor GHGs and aerosols. Does its climate model also assess different sources of climate sensitivity? It seems like the

modal forcing in the present study is slightly higher than in Rennert et al. Is that difference due to accounting for aerosols? If so, this would be very interesting in light of the recent public debate (featuring Hansen, Mann, and others) regarding the role of aerosol pollution reduction in the future warming outlook. Please elaborate on these differences and their causes, as they are potentially an important contribution of the paper.

We recognize that the underlying methodology is quite parallel to the Rennert et al. paper. We have made advances through the use of the Silicone model and the use of an updated version of the FaIR model. However, the more important advance of this paper is the recognition of the importance of the radiative forcing results for guiding future scenario selection – Rennert et al. could have made a similar conclusion based on their own modeling, but their focus was on their method’s usefulness in the SC-GHG context instead of the scenario design community.

3. The paper should discuss its ‘probabilistic’ projections a bit more circumspectly. The Rennert/Raftery/Müller et al./RFF methods for probabilistically projecting emissions and their drivers are the best we currently have, but assessing probabilities for these quantities out to 2100, and even more so to 2300, is still extremely fraught and subject to deep and unquantifiable uncertainties. I don’t think it is defensible to take such probabilistic projections literally, even though the density of scenarios does provide some valuable information that is partly related to probability.

We recognize that no one approach produces “truth” regarding future uncertainty, particularly when it comes to socioeconomic questions. We try to appropriately caveat our assumptions here. E.g., “Of course, projecting even to 2100, less than 80 years into the future, is a challenging task, particularly for socioeconomic conditions that are key factors in determining future emissions. Barron (2018) discusses some of the challenges in projecting future penetration of low-carbon technologies using climate policy models. While the RFF-SP scenarios may be the best probabilistic emissions forecast available, these probabilities must be taken with some skepticism, particularly in the tails (such as the 1% threshold that we focus on).” We added the sentence, “Projecting socioeconomic conditions and emission drivers through 2300 is an even more challenging task, and yet is necessary for considering the long-term implications of long-lived GHG emissions” to further emphasize the difficulty (and yet necessity) of these kinds of calculations.

4. Relatedly, the extensions of socioeconomic scenarios from 2100 to 2300 used in this study are extremely speculative. This should also be acknowledged.

Again, we recognize the challenge of projection socioeconomic factors to 2100, much less 2300 – unfortunately, projections of these time periods are necessary for certain applications, and therefore the attempt is still worth making. We attempt to include sufficient caveats (see above).

5. Lastly, to the extent space allows, it would be helpful to discuss the mechanistic aspects of the RCP8.5/SSP5-8.5 debate a bit, in context with the model’s results. Probabilistic projections based on historical trends, such as from RFF, etc., are valuable, but should also be discussed in context with mechanistic questions about the plausibility of RCP8.5/SSP5-8.5’s coal assumptions (see Ritchie and Dowlatabadi 2017 Energy 140, 1276, Pielke and Ritchie 2021 Energy Research & Social Science, 72,

101890, and Burgess et al. ERL 2020 already cited), SSP5's economic assumptions (see Burgess et al. 2023 Communications Earth & Environment 4, 220), SSP5-8.5's land-use assumptions (see Schwalm et al. 2020, already cited, and Hausfather and Peters 2020 PNAS 117, 27791), etc.

The RFF-SP projections are based on expert elicitations, not historical trends. Therefore, the method should appropriately capture the plausibility of the underlying socio-economic and emissions projections. Remember that an important distinction between our approach and the approach of Ritchie, Pielke, Burgess, Schwalm, and Hausfather is that our approach includes both climate uncertainty and socioeconomic uncertainty when looking at the plausibility of reaching 8.5 W/m^2 in 2100, whereas the other papers all concentrate only on the emissions. Only 3 out of 10,000 RFF-SP scenarios have emissions as high as SSP5-8.5 in 2100: this seems to be appropriately low probability. In contrast, 53 of the 10,000 RFF-SP scenarios exceed 8.5 W/m^2 in 2100, showing that the probability of radiative forcing exceedance can be substantially larger than the probability of exceeding the emissions considered in earlier literature.

Specific comments:

Line 53: "RCP8.5 has been characterized as a scenario that assumes high population growth, low income growth, and modest improvements in energy intensity over the next century". The authors should note that these population growth and income growth assumptions are reversed in SSP5-8.5 (very high income growth and low population growth), and the high income growth assumption is key to the energy demand in SSP5-8.5; see Riahi et al. (2017).

We have added this clarifying sentence: "Meanwhile, SSP5-8.5 followed a different socioeconomic path relative to RCP8.5, with low population growth coupled with high income growth that drives energy demand, demonstrating that the same forcing scenario can be reached by very different pathways."

Line 98: Define 'GWP' (global warming potential, I think, but the same acronym is also commonly used for gross world product) at first mention.

Added definition at first mention.

Line 116: There are a few other rationales for using (or for not using) high-emission scenarios that have been proposed. Some are reviewed by Burgess et al. (2023) ICES Journal of Marine Science 80, 1163.

We agree with the conclusion of Burgess et al. to not refer to 8.5 as a BAU scenario, and the importance of clarity for the impacts community so that, for example, adaptation decisions are not being made on the basis that 8.5 W/m^2 is a likely outcome. However, as noted above, the literature reviewed by Burgess are all focused on the plausibility of the emissions and socioeconomics in SSP5-8.5, which is a different question from the plausibility of an 8.5 W/m^2 radiative forcing scenario.

Line 142: Deep uncertainty and discounting also affect the policy relevance of warming that may occur in 2150 and beyond. Analogously looking back to 1890, it is difficult to imagine how many of today's

policy challenges would have been predictable to the people then—including the climate challenges and technologies available for adaptation and mitigation. The same goes—even more so—for 2300.

We agree with the deep uncertainty involved in projections through 2100, much less through 2300. However, these projections are necessary for some applications (such as the SC-GHG), and therefore it is necessary to make some attempt to make best estimates for these time periods (with appropriately large uncertainty bounds).

Line 146: There is a rich literature in economics on how decision-makers should handle low-probability, high-impact events. I suggest looking at and discussing Weitzman (2014) *American Economic Review* 104, 544, Costello et al. (2010), *PNAS* 107, 8108, and Anthoff and Tol (2022) *Journal of the Association of Environmental and Resource Economists*, 9, 885.

We recognize that there is literature regarding low-probability high-impact events. We are familiar with how net present impacts can explode when there are fat-tails in the future damage probabilities. However, we note that our probability functions here do not have fat tails. We looked for literature from probability & risk experts such as Granger Morgan and Baruch Fischhoff, but have not yet found any papers which quantify the appropriate probability threshold to use for a low probability/high-impact scenario beyond discussing the relationship between the higher the impact the lower the acceptable probability.

Line 194: To what extent does the paper’s ensemble capture uncertainty in climate-system feedbacks? This is a possibility often discussed by proponents of using RCP8.5 as a prominent scenario.

The FaIR model includes carbon cycle and ocean heat uptake responses to climate change, as well as climate sensitivity uncertainty, all of which are key feedbacks. This is part of what increases the probability of reaching 8.5 W/m^2 by the end of the century. We state that, “any feedbacks that don’t exist in either the historical record or the climate models against which FaIR was calibrated will not be captured by this analysis. For example, the permafrost feedback is not included in FaIR, which has the potential to contribute substantially to radiative forcing and warming through release of additional CO_2 and CH_4 ”

Line 266: This isn’t necessarily true if the timing of the warming matters to the impact.

The use of by-degree/global warming level/time-shift/pattern-scaling approaches mean that the impact modeler can disassociate the climate model output from the climate model time period. E.g., if a policymaker cares about a 2 degree stabilization scenario, then the 8.5 W/m^2 data from the decade where 2 degrees is reached can be used as a proxy for the 2 degree outcome in 2100. We have added text to clarify this: “However, this does not mean an 8.5 W/m^2 scenario can’t be relevant: the use of a by-degree approach allows for disassociating the climate model output from the climate model time

period. This approach – referred to as “time-slices” (Schleussner et al., 2016), “time-shift” (Tebaldi and Knutti, 2018), or “temperature binning” (Sarofim et al. 2021) - allows for the GCM output of early decades from a high emissions scenario to be used as analogues for temperatures in later decades of lower emissions scenarios. Reduced complexity approaches (such as the combination of FaIR and the RFF-SPs) can be used to determine the arrival times of these temperatures: moreover, these approaches can also produce the probabilities of exceedance of different temperature thresholds, which would be useful information for adaptation decisionmakers and other policymakers."

Line 274: It is not clear what on what basis this claim is made or on what basis future evidence could reverse it. What probability threshold is being assumed?

The relevance threshold used in this paper was 1%, which is substantially exceeded by 2150. We have updated the final paragraph as follows:

While we argue that 8.5 W/m^2 continues to be relevant based on our probabilistic analysis, we can hope that as zero carbon technologies continue to become more competitive, and mitigation efforts continue to increase, the 8.5 W/m^2 scenario may eventually drop enough in likelihood that it would no longer exceed our 1% threshold in any year and could be discarded.

Reviewer #3 (Remarks to the Author):

A review of the manuscript entitled “How high is high enough? A multi-million-member ensemble analysis of future climate scenarios and their relevance” by Sarofim et al.

The authors used a 10 million-member-ensemble plausible future realizations to examine the possibility of exceeding 8.5W/m^2 in a future warming climate. They found that the probability of exceeding this threshold in 2100 is less than 1%. Though with an extremely low probability, the authors argued that it is still of societal relevance for the community to continue to use a scenario that reaches 8.5W/m^2 in 2100 for serving as an analogue for post-2100 climates and developing damage functions that can be useful through 2300. The results, however, are not surprising and do not represent conceptual novelty over previous studies. I have several concerns about the interpretation and presentation of the results, which preclude me to recommend acceptance of the paper in its current form for Nature Communications.

Major comments:

(1) Lack of novelty. The main idea in this study that it is extremely unlikely to exceed 8.5W/m^2 by 2100 (i.e., equivalent to RCP8.5 and SSP5-8.5 scenarios) is essentially the same as Hausfather & Peters (2020), except that the present study uses a super large ensemble with multiple members and includes some analyses of uncertainties. However, the uncertainty part (Fig.3) is also a kind of repetitive analysis of Hawkins and Sutton (2009) and Lehner et al. (2020), both of which already concluded that the uncertainty due to internal variability (similar to the climate parameter uncertainty used in this study;

Fig.3) is a main driver of near-term climate.

The evaluation of Hausfather & Peters (and other studies cited in our paper) were focused only on emissions. Moreover, they were not probabilistic analyses. This paper resolves those limitations and is therefore an important advance relative to this earlier literature.

From Hausfather & Peters: “some important feedback effects — such as the release of greenhouse gases from thawing permafrost — might be much larger than has been estimated by current climate models” and “Those who are tasked with taking climate action on the basis of information from model scenarios are increasingly calling for a more risk-based approach to help with adaptation and mitigation. This approach accounts for the relative likelihood of different outcomes. *Controversially, it requires researchers to assign probabilities to scenarios*” (emphasis added).

The IPCC (Chen et al. 2021) stated that “On the other hand, the default concentrations aligned with RCP8.5 or SSP5-8.5 and resulting climate futures derived by ESMs could be reached by lower emissions trajectories than RCP8.5 or SSP5-8.5. That is because the uncertainty range on carbon cycle feedbacks includes stronger feedbacks than assumed in the default derivation of RCP8.5 and SSP5-8.5 concentrations.”

Similarly, a limitation of Hawkins & Sutton and Lehner et al. is that neither the underlying emissions scenarios nor the climate models informing their analysis were probabilistic, but rather ensembles of opportunity where the individual members were presumed to be equally likely. Only the internal variability uncertainty was properly quantified. Therefore, their comparison of different uncertainties was not really apples to apples. Again, this study is an important advance relative to these previous studies.

From Lehner et al: “Scenario uncertainty can be quantified by comparing a consistent and sufficiently large set of models run under different emissions scenarios. This uncertainty is considered irreducible from a climate science perspective, as the scenarios are socioeconomic what-if scenarios and do not have any probabilities assigned (which does not imply they are equally likely in reality).” And “Another important source of uncertainty not explicitly addressable within the CMIP context is parameter uncertainty. Even within a single model structure, some response uncertainty can result from varying model parameters in a perturbed-physics ensemble (Murphy et al., 2004; Sanderson et al., 2008). Such parameter uncertainty is sampled inherently but non-systematically through a set of different models, such as CMIP.” And “Note that the assumption of symmetry is an approximation, which is violated already by the skewed distribution of available emissions scenarios (e.g., 2.6, 4.5 and 8.5 W m⁻² in CMIP5) and possibly also by the distribution of models, which constitute an ensemble of opportunity rather than a particular statistical distribution (Tebaldi and Knutti, 2007). Thus, the figures corresponding to this particular calculation should only be regarded as an illustration rather than a quantitative depiction of the multimodel multi-scenario uncertainty. Also, the original depiction in HS09 was criticized for giving the impression of a “best-estimate” projection resulting from averaging the responses across all scenarios. That impression is false since the scenarios are not assigned any probabilities; thus their average is not more likely to occur than any individual scenario.”

(2) Insufficient explanation of physical mechanisms. The authors spent a large part of text on “Discussion”, which, however, only focused on “implication” rather than the “cause” of the results. For example, while the climate parameter uncertainty and emissions uncertainty equally contribute to total forcing uncertainty by 2050, climate parameter uncertainty exceeds emissions uncertainty almost though 2100 for temperature uncertainty. This is an interesting result of great importance but the authors failed to explain why, for instance, by which physical processes and/or through which climate variability modes this would occur. The provided explanation (Line198) is rather mathematical than physical.

We discuss key climate parameters in FaIR in the following paragraph: “This is a result of fewer uncertain climate parameters that impact radiative forcing (e.g. aerosol forcing, carbon cycle, and methane feedback uncertainties) than impact temperature (e.g. all the parameters that impact radiative forcing also matter for temperature, but uncertainties in climate sensitivity and ocean heat uptake also have substantial temperature impacts).”

We don’t do an explicit decomposition of parameter influence, as that would require substantial additional analysis holding each parameter constant in turn, but based on conversations with the developer as well as other research in this area (e.g., work by Chris E Forest in the 2000s with the MIT IGSM), generally the dominant uncertain parameters for future temperature are climate sensitivity and ocean heat uptake, and for radiative forcing, ocean carbon uptake.

(3) Ambiguity of methods. The definition of the three types of uncertainty is difficult to understand. For example, what do climate parameters specifically represent? Internal variability? Or model physics that determine certain oceanic and atmospheric feedbacks? Or others? Which climate parameter is used in calculating emissions uncertainty? Which single emissions scenario is used to compute climate uncertainty? There lacks clear step-by-step description of procedures and associated rationales for calculating the uncertainties.

We have improved our discussion, graphical representation, and calculations of the differentiation between stochastic internal variability (an aspect of the model which can be turned on or off), climate model parameters (e.g., ocean heat uptake, carbon cycle parameters, climate sensitivity), and emissions uncertainty (based on the RFF-SP for socioeconomic parameters).

(4) Related to my concern above, to what extent does internal variability contribute to each of the three uncertainties? what is the role of internal variability in partitioning/affecting the uncertainty contribution? Why the uncertainty range (gray shading) initially increases, then peaks around 2100 and slightly decreases thereafter in CO2 emissions (Fig.2), while the uncertainty range continuously increase through 2300 for radiative forcing (Fig.3)? In Fig.3a, after ~2080, emissions uncertainty nearly accounts for 100% of total combined uncertainty, while climate parameter uncertainty remains large (>40%), how to interpret this? In Fig.3b, why in ~2030 the fractional contribution of climate uncertainty can even exceed 100%?

We have separated out internal variability as a separate measure, and changed our figure display for Figure 3 to improve clarity.

(5) The authors noted that internal variability contributes to all the three uncertainty distributions. However, the internal variability may be vastly different in each uncertainty, making it impossible to compare the two emissions and climate uncertainties. For example, at a certain year when emissions uncertainty exceeds climate uncertainty, internal variability may play a bigger role in the emissions uncertainty. As such, when the influence from internal variability is excluded for both cases or exactly the same internal variability is used for each case, the “pure” emissions uncertainty may not exceed climate uncertainty, which undoubtedly undermines the robustness of the results and conclusions presented in this study.

We have separated out the internal variability uncertainty in order to clarify these differences.

(6) Line 210: as noted by the authors, there are many other different approaches in developing emission scenarios. Why the authors choose to use RFF-SP and consider the RFF-SP to be the best? What is the biggest advantage of RFF-SP?

We added additional text to the methods stating that: “RFF-SP is recent, open-source, fully probabilistic, extends through the year 2300, and is used by the U.S. SC-GHG process, making it an ideal dataset for emission projections.”

In an ideal world with infinite resources, we would repeat this analysis with a second set of uncertain emissions scenarios to provide an estimate of structural uncertainty in emissions projections in addition to parametric uncertainty (we could also look at structural uncertainty in simple climate models by running MAGICC in parallel to FaIR). However, even if we had those resources, we are not aware of other probabilistic emission scenario ensembles that extend through 2300, and few that extend to 2100.

(7) For the second application, the authors argued that climate projections for 2100 under high forcing scenarios can be used as analogues for climates that may not be seen until much later (Line 135). However, climate impacts are not linearly related to the intensity of forcing scenarios and there are hysteresis-like behaviors to CO₂ forcing. That said, climate projections for 2100 cannot be used as analogues for climate in 2200, even under the same high forcing scenarios. In this case, the associated impacts and mitigation and adaptation strategies are also completely different.

There are two different issues being raised in this comment: the first is what climatic differences are there between 3 degrees in 2100 and 3 degrees in 2200 (recognizing that FaIR does address inertia & hysteresis in the relationship between forcing and temperature). Other authors (e.g., Tebaldi et al) have concluded that while there will be some differences (e.g., since land warms faster than ocean, a scenario which reaches a 3 degree global average in 2050 will likely have warmer land and cooler oceans than a

scenario which reaches 3 degrees in 2100), these differences will be small relative to other uncertainties.

The other issue is about the impacts, mitigation, and adaptation strategies differing between 2100 and 2200. The by-degree approach has no problem with this: the application of the time-shift/time-slice/by-degree approach involves applying the temperature, precipitation, and other physical factors from a three degree climate in 2100 to a socioeconomic world in a year of the analysts choice (2050, 2150, etc.).

Specific comments:

(1) Line61: what is BAU?

Clarified for first usage of BAU

(2) Line98: what is GWP-weighted?

Clarified in a footnote to Table 1 (now the first usage of GWP). Additionally added in first use in main text.

(3) Line104: should be Table2?

Corrected

(4) All the probability presented in Table1 and 2 should have corresponding uncertainty ranges. The question of whether uncertainties should themselves have uncertainties is an interesting philosophical discussion (e.g., does it make sense to say that there's a 20% (10%-30%) chance of rain tomorrow?), but my impression is that most of the community does not present ranges for uncertainties. We have improved the titles and captions for the tables and figures to be more precise, as these are actually calculations of the percent of scenarios that exceed a given threshold, which is a perfectly known number (as versus the actual probability of exceedance).

(5) Line150: delete "to be".

The sentence has been revised to read more cleanly: "The question then becomes at what likelihood does a scenario become relevant"

(6) Line159: what is the rationale for choosing 1%? It seems a very arbitrary choice.

We agree that the choice of a 1% threshold is somewhat arbitrary – as we stated in response to Reviewer 1, we looked for literature from probability & risk experts such as Granger Morgan and Baruch

Fischhoff, but have not yet found any papers which grapple with the problem of exactly how low a probability is relevant.

(7) Table1: what does "GtCO₂e" represent?

Clarified in table footnote

(8) Fig.2: add units in y-axis.

Added

(9) Fig.3: add labels "a, b, c, d" in each panel.

Added

Reviewer #4 (Remarks to the Author):

The authors aim to assess The likelihood and relevance of using RCP8.5, which may be relevant for the scientific community using scenarios for climate and adaptation analysis.

Thank you

The Abstract and Introduction appear unstructured and confusing. However, with a few changes, you could appear engaging and provide an overview of the relevant context and knowledge gaps and what is intended by the authors (which is now more or less missing).

The introduction and abstract have been substantially revised to be more clear

The paper needs more focus and a major rewrite of the abstract and introduction), and I think the authors write to a very narrow audience (both writing style/explanation level and focus). You could expand and include all your results rather than only focus on 8.5.

While we still feel like 8.5 deserves special focus because of the upcoming ScenarioMIP choice of highest scenario to use for CMIP7, we have expanded our discussion of other results as well.

I got confused about which scenario series you use (be clear from the beginning) and only realized it in the results/Table 1), so that reflects my comments up till then. I probably made several repetitions in the comments ;)

We have tried to be more clear about scenario choices.

Abstract:

You could improve the abstract by first presenting the context you are writing into (which area of interest) and the knowledge gap (why is this study important), maybe the method/approach, results,

and finally, what the results contribute to in the abstract.

Right now, it is a bit messy ;)

Decide if you wanna use the term emission or forcing (make it simple to read)

The difference between high-emission and high-forcing scenarios is an important part of this paper: we have tried to be clear as to which is which.

You write "calibrated to match the IPCC 6th Assessment Report." What does this mean?

Clarified: "whose uncertain parameter distributions are consistent with assessed uncertainty ranges from the IPCC 6th Assessment Report". In the methods, we also elaborate, "The FaIR calibration we use (v1.0; Smith 2023) is consistent with the IPCC AR6 Working Group 1 assessment of present-day warming, equilibrium climate sensitivity, transient climate response, present-day aerosol radiative forcing, present-day CO₂ concentrations, and recent-past ocean heat content change, including the uncertainties in these distributions (Forster et al. 2021; Smith et al. 2021), maintains a root-mean-square error between modelled and observed global mean surface temperature of less than 0.16°C in each ensemble member, and is consistent with historical emissions from RCMIP."

I suggest you add more context, e.g., state why you are doing the study, why it is important, the knowledge gap, and finally, what the results contribute to in this larger context.

We have attempted to clarify, in particular by citing the need to choose an upper bound scenario for the newest ScenarioMIP climate scenarios.

Introduction:

The same comment goes for the introduction. Right now, it is a bit messy, and you can improve by introducing the greater context and then going more into detail. And engaging the reader (e.g., introducing RCPs 8.5 as "the highest of four concentration scenarios" could come in sentences 2,3,4 or 5)

Thank you for the suggestion: we now introduce RCP8.5 in the first sentence, and have deleted some extraneous material.

I prefer more context to inform (and engage) the reader. I am not sure, but your study objects are the RCPs 4.5 and 8.5. Why are the RCPs interesting/important other than being used for AR5? They were published more than ten years ago. They may still be important for analyzing future CC, and thus, high-impact (low probability) scenarios are important for adaptation analyses.

We clarify that we are looking at 8.5 W/m² as inspired by both RCPs & SSPs.

I think the context (as you mention later) could be the discussions about the relevance of RCP8.5, which is the discussion you seek to participate in - the relevance of your paper's theme. The key or initial paper would be Hausfather/Peters (2020): <https://www.nature.com/articles/d41586-020-00177-3>. Check also Pedersen et al. (2020): <https://www.nature.com/articles/d41586-020-00177-3>.

We have added a reference to Pedersen et al. (we already referenced Hausfather).

Reading the Intro, I'm confused about your focus. Are you focusing on RCPs (4.5 and 8.5) or SSP-RCPs? Your research would appear more up-to-date if you assess the SSP5-85 (less likely) instead and compare it with the SSP2-4.5 as "the current policy" case. The SSP3 is also a high-emission scenario, and the SSP3 narrative is quite similar to recent trends with national conservative moments, increased national security, and national defense budgets.

We have attempted to clarify.

I. 416: Seeing Table 1, I realize which scenarios you are examining. Make it clear what you are focusing on earlier :)

We have attempted to clarify.

Why not assess the entire suite from 1.9 to 8.5 if you focus on forcings? It would be highly interesting to know but maybe a lot of work! At least the upper end, intermediate (4.5), and lower ends (1.9 and 2.6) would be interesting also from a policymaker perspective.

Table 2 shows all the scenarios from 2.6 through 8.5, and we have added more discussion of the scenarios other than 8.5.

I'm still not convinced it is relevant with one more paper assessing 8.5, but maybe I'm wrong. I suggest that you focus on the entire suite and frame it more broadly: can we reach the Paris Agreement, or where are we heading, and can we go as high as 8.5? (this is just a suggestion to make it more policy-relevant and relevant for a larger community. A 70% chance of exceeding 4.5 and 90% exceeding 3.4 is quite interesting, and stating/confirming (SSP2-)4.5 as a "current policy" case. I would change your scope/context :)

Because of the upcoming decision in ScenarioMIP about the highest scenario to include for CMIP7, the question of whether to include 8.5 is very immediately relevant. However, we have added more discussion regarding other scenarios along with median results.

In addition, many have assessed the likelihood of these scenarios and the earlier generation. You would show a convincing overview if you justify why you assess "probabilistic emission projections" on scenarios that are not probabilistic or based on likelihoods. Historically the role of scenarios (probabilistic or qualitative scenarios) has been discussed since Schneider (2001) vs. Nakicenovic (2002) - but easiest to mention that there is an overview of this in this 2022-paper:

<https://www.sciencedirect.com/science/article/pii/S0959378022000760>.

We appreciate the Pedersen et al. citation, it is a great summary of this topic area. Note that the lead author of this paper (Sarofim) was an author on Webster et al. 2002, which was cited by Pedersen as an early entrant into the probabilistic v. storyline approach to scenario analysis... and I (Sarofim) still believe in the value of estimating the likelihood of exceeding the key marker scenarios (Hausfather & Peters, which you cite, also talk about the value of requiring "researchers to assign probabilities to scenarios")

What does this mean: "given the latest available societal projections."

Clarified to, "given a recent probabilistic ensemble of socioeconomic emissions"

The method description, 83-92, is quite nice. But be sure to explain what some acronyms stand for and mean as a method, e.g., RFF-SP.

Defined RFF-SP acronym.

Results:

I am not sure what is RFF-SP scenarios: “We first compare emission levels across the RFF-SP scenarios to those in the SSP scenarios” (l. 95)

This needs to be explained better.

We have improved our initial introduction of the RFF-SPs: “First, we use a recently developed ensemble of probabilistic emission projections (the Resources for the Future Socioeconomic Projections, or RFF-SPs, Rennert et al. 2022a, Rennert et al. 2022b)” and added some other elaborations about the RFF-SPs in other locations.

For me, the results are written to an even narrower audience. I would expect at least an explanation of what the results and numbers mean (l. 100). It's like you have been writing to your colleagues ;) Practice communicating it to a broader audience and explain what the numbers mean :)

Anyway, explain the results in a few lines. In my opinion, it is not enough to present them in a table (l. 100), although you discuss them later :)

We have added more discussion of the results from tables & figures in the results section

Which “six SSP scenarios” are you analyzing? (l. 97) You would benefit from being more informative about what you are doing and what you are studying and why Introduction to results

We now list the scenarios in the sentence: “six most widely used SSP scenarios (SSP1-26, 4-34, 2-45, 4-60, 3-70, and 5-85)”

In essence, results and discussion works better, are more stringent and less messy, and have better quality than the Abstract and Introduction. Your conclusions about the “probability of exceeding this threshold in 2100 are less than 1%, it reaches 7% by 2150, and 20% by 2300” are interesting, but SSP5-RCP8.5 is mainly interesting related to adaptation assessment as the high-impact (low probability) case. You, of course, add some value here and more numbers, but how does your work differ from, e.g., Christensen's (I think they focused on GDP, right?) - be specific about this in the introductory (which context you are writing into and what your results contribute to). As stated earlier, I would create a new focus using all your results and not focusing only on 8.5.

In this paper, we have tried to show why SSP5-85 is of interest to the pattern-scaling community and social-cost-of-carbon community.

Christensen et al. is in fact one of the few probabilistic assessments of end of the century emissions. However, they rely on the DICE model for the carbon cycle calculations, and FaIR is a substantial improvement in that area (see discussion in the NAS climate damages assessment) (I am also surprised that the inclusion of carbon cycle uncertainty made such little difference in Table S3 of Christensen et al., though I think it is easy to overestimate how much uncertainty will increase when combining multiple distributions... our own Figure 3a shows a similar effect). Perhaps more importantly, Christensen et al. is 5 years old, and there have been substantial advances in mitigation in those 5 years which have been reflected in the RFF-SPs that have substantially reduced the probability of higher emission scenarios.

Detailed comments:

Introduction: create a new beginning where you include the overall context and rationale of the paper :)
E.g., don't begin with the RCPs if you are examining the SSP-RCPs

We have rewritten the introduction to discuss the SSP-RCPs in the same first sentence as the RCPs

I. 41: US - write the United States for the first time

We no longer reference this

I. 41-46: Is this an important contribution to the science to mention this as a backup for your choice of scenarios? (I don't know the reference, so I will not define if it is or not) "and the US 42 National Climate Assessment (NCA4) and innumerable research papers. While NCA4 used the full 3 43 range of RCP scenarios, the assessment focused most closely on RCP4.5 and RCP8.5, which they 44 considered as low-end and high-end scenarios, to be consistent with the range of scenarios used in 45 previous assessments (USGCRP, 2015)."

We have deleted this reference

I.83-87 becomes quite technical. I would introduce and explain what RFF-SP is in more detail (1-2 sentences) before using it in an RQ

I.95: "We first compare emission levels across the RFF-SP scenarios to those in the SSP scenarios"

Explain what this means for the less technical reader. Again, at this stage, RFF-SP still appears relatively unknown to the reader, and be more precise: are you assessing the SSP-RCPs via a method, or are the RFF-SP different scenarios (I guess the first, but I'm not sure actually)?

RFF-SPs are their own set of probabilistic emission scenarios: we have tried to clarify that in the text.

REVIEWER COMMENTS

Reviewer #1 (Remarks to the Author):

The authors have attempted to address the comments raised in the previous round. By sending the paper back out for review, the editors seem to have made a decision about the paper's novelty being sufficient for publication. If that is indeed the case, I suggest providing the authors with one more chance to revise. In their revision, I encourage them to consider the following small but important points:

1. Probability: I still think the framing of the results as probabilistic deserves at least a caveat at the beginning, in addition to the one added to discussion. The RFF framework does not capture all possible sources of uncertainty, even though it tries to capture the range of expert opinion about uncertainty. These two things have sometimes been drastically different from each other, for example in the years leading up to the 2008 financial crash, or in advance of the COVID-19 pandemic.

2. In Table 1, the exceedance probability is sometimes the same or higher for SSP4-3.4 as it is for SSP1-2.6. I think this is because the emissions trajectory for SSP4-3.4 dips slightly below SSP1-2.6 right before 2100, but to readers who don't know the SSPs well, this will seem like a very strange result (because 4-3.4 is a warmer scenario than 1-2.6). Therefore, I suggest explaining it.

Reviewer #3 (Remarks to the Author):

The paper now reads better after clarifying many issues raised by me and other reviewers. My comments in this round are as follows.

1. The authors state that 53 of the 10,000 RFF-SP scenarios exceed 8.5 W/m^2 in 2100, showing that the probability of radiative forcing exceedance can be substantially larger than the probability of exceeding the emissions considered in earlier literature. What causes the difference between the probability of radiative forcing exceedance and emissions exceedance? Can you elaborate on this point and give some explanations in more detail?

2. The uncertainty part is crucial to the results but the authors didn't positively address much of my concerns in my previous comment #4. My further question is from your new Fig.3, can we conclude that emissions uncertainty is the largest source of uncertainty to climate model projections as greenhouse warming continues?

3. As the core of this study is based on the simple climate model FaIR, you'd better provide some evidence showing that the model is reliable and good at capturing observations, especially to people outside of the field and not familiar with the model and methods.

4. Fig.1: the colors used for now are extremely difficult to distinguish different SSPs

5. Line106: should be "SSP3-7.0"

6. Line148: with an 88% chance of exceeding 2oC in 2100?

7. Line377: what does the 50th percentile mean?

8. As acknowledged by the authors that individual uncertainty components may exceed the total uncertainty, the use of limit to 100% in y-axis in Fig.3 is not appropriate and misleading.

Reviewer #4 (Remarks to the Author):

Nice improvements. Well done. Reads better now. I like the paper and enjoyed reading it. It is almost done, in my opinion, but there are still some issues to solve :)

I made several repetitions - I didn't delete them all, so reply once to each point ;)

I like the introduction better, and it focuses on adding probability to the scenarios. Just remember that the RCP-SSP scenarios were never intended to be based on frequency distributions or probability analysis—they are all plausible scenarios without probability attached. So it is actually great that you question why the RCP modelers added probability to 8.5 as the only scenario—maybe explain better how they should have done it, in your opinion.

Make it clear in the introduction or results that you compare two different types of scenarios. It could be something like the following:

The RFF-SPs are probabilistic emission projections, forecasts about future greenhouse gas emissions (and radiative forcings) that account for uncertainty, e.g., presenting a single (best-guess??) prediction including a range of possible outcomes or 82 probabilistic emission predictions. Forecasts often include predicting likely futures based on frequency distributions. these are compared to the RCP-SSPs and RCPs scenarios. The RCP-SSPs are different from the RFF-SPs. They are qualitative descriptions of alternative futures. They are not forecasts or predictions of most likely futures but descriptions (or projections) of alternative futures that are plausible but not necessarily probable. They are typically based on different sets of assumptions about how various factors might evolve over time. Scenarios are used to explore the range of possible outcomes and to understand the potential consequences of different decisions or events (e.g., Webster et al., 2003; Pedersen et al., 2022).

In the abstract, I would state that the low end of the SSP-RCP range is within the range of your results, while SSP5-8.5 is outside the upper end. That way, your results are presented in a more complete and balanced way, and it highlights your aim: to discuss SSP5-8.5.

Focusing on (the low probability of) RCP8.5 is relevant; however, it has always been a low-probability, high-impact case aiming to inform the worst possible outcome for adaptation analysis (as you also more or less address). I would be surprised if it will be excluded from the upcoming SSP elaborations or CIMP for the AR7 analysis. I think your introduction puts it much more to the point with the statement from the ScenarioMIP workshop report.

I'm curious about your model and the probabilistic analysis beyond 2100. On which basis did you develop the SSP extensions beyond 2100, e.g., what are the underlying assumptions (e.g., mathematical, socioeconomic)? In particular, they are relevant to discuss because 7.0 and 8.5 start tracking downward quite fast after 2100. Why? This info is missing in the methods (I think?). You will demonstrate a good understanding and critical self-evaluation if you elaborate on why you track down this fast in the discussion section and also discuss the reliability of tracking downward at a relatively fast speed.

Does your "mean" estimate represent a best-guess scenario? I'm curious. You mention the concept but don't say whether you are presenting one. Between RF 4.5 and 6.0 is not far from most assessments of current policies and/or NDCs.

To sum up, the most important issues to resolve and clarify: ^[11]_[SEP] why does the paper discuss BAU and best-estimate (intro and discussion); and if it presents a best-estimate scenario or not (and why or why not if it does not present one)

- Why SSP5-8.5 and SSP3-7.0 track downward rather fast beyond 2100 - some mathematical assumptions and a critical discussion about the reliability in a real-life world of this to happen. I'm not saying it is wrong, but I would expect some critical self-reflections - it would be interesting and valuable with a reliability discussion both within the model and its inputs and between the model and the real world. In essence, how do your model or assumptions explain the SSP behaviors post-2100?

- Clearly state the differences between the scenarios you compare, how are they comparable, and the conditions for comparing to different (sets and) types of scenarios. The RFF-SR are probabilistic emission projections, while the SSP-RCPs are not. It's fine to compare them, but it is essential to state the differences and the conditions for comparisons (introduction or results and discussion, and maybe methods).

- A conclusion where you summarize the results and long discussion and clearly state your recommendation (opinion) - not everyone will read the entire discussion.

- The modelers never intended the BAU concept. It was labeled like that by several scenario users. It is ok to discuss it but don't make the same mistakes as others... in particular if it is part of your justification to examine 8.5 - check I think Pedersen et al. 2021 or 2022 (one of these presents the historical use of BAU since the first IPCC scenarios in 1990 and how modelers started to use a DAU concept instead - but apparently the users continued with BAU :) - the 2021 paper looks at scenarios vs. history and must be easy to find.

Why did you choose the Rennert scenarios? e.g. What are the advantages? (In "Methods," add a line in the nice paragraph explaining the scenarios in the "introduction.")

Abstract:

It's quite improved and nice, with the final putting the results into perspective. However, it's still quite technical, which narrows your reader group. I would prefer you begin with 1-2 sentences comprising some context and introducing the field and problem you wish to address.

Less technical language could invite readers who are normally far from or outside the statistical field to understand how our results are useful. For example, damage function could be swabbed with or supplemented by impact or adaptation analysis. This may bring your writing closer to the real world and maybe also inform decision-making or disaster-planning actors.

Introduction:

Very improved introduction. I think you will catch your audience much better here and introduce your aim well.

Line 50 (+ Discussion 276): I would modify this. RCP-85 was never intended to be a BAU; however, it was used by several scenario users as such. This was explained in one of the Pedersen et al. papers, maybe in 2021 or 2022 (IPCC mandated a stop of this term, so modelers started using dynamics as usual instead). You could write that it was mistakenly perceived as a BAU by some scientific scenario users but was never intended as such by the developers. As I remember, Hausfather/Peters (2020) misinterpreted it a bit - which happens, but don't do the same ;)

Line 51: I'm not entirely certain that your RFF-SP represents a best-estimate scenario. Could you clarify whether it does or does not?

Lines 80-90. I like the inclusion of this paragraph—it supports reading the results and can be improved further, E.g., clarify what variables the analyses are based on. Is the statistical analysis based on emissions and forcings alone? Or does it also include other variables like economy, policy, and population (which are included in the SSPs and SPAs (and SSP-RCPs))?

You could additionally write in a few sentences clarifying for non-statisticians what the probability assessment covers in a non-technical language.

In essence, I would improve the quite nice and short description (lines 80-90), opening up for a better understanding of the rather abstract and less transparent functions. It is distracting to read the results when I am unsure about what the probability is based on (it would be nice to know what is lying behind)

It could be something like:

The RFF-SR are probabilistic emission projections or forecasts about future greenhouse gas emissions that present a single prediction accounting for uncertainty or ... instead of a single prediction, they provide a range of possible outcomes. They are based on the Future Socioeconomic 82 Projections (RFF-SPs), developed by Rennert et al. (2022ab), based on ...

(Just ideas, use your own words to make it transparent what variables the analysis is based on, Kyoto GHGs or also other variables - I have not read the methods yet). After reading the entire paper, I would suggest you summarize the main characteristics of the Rennert scenarios and your model. Make it more transparent what your scenarios represent - they are more than numbers and equations, open them up for the reader so we can better evaluate/interpret the results

Line 39-40: “Shared Socioeconomic Pathway (SSP) 5-8.5 for CMIP6”. I would write the RCP-SSP combination and include Gidden et al. (2019)

Lines 54-56: references seem to be missing

Lines 86-87: Are any references needed here?

Line 88: “and other dates”. Dates or years? Why not write the exact years?

88 -90: Belongs to results

Results:

Interesting results, both emissions and RF.

The findings would appear more trustworthy for non-statisticians if I understand better what the RFF-SP probability expresses or is based on (i.e., include in the description in lines 80-90)

Fig. 1: Nice figure btw :)

What data source did you use for the RCP-SSPs in Fig. 1? If you didn't generate the SSP data yourselves, I would add that source in the figure text (for the SSP database, Riahi et al., 2017 will be sufficient).

NB: After reading the methods, I see that you extended the SSPs. Please clarify what data you used and what is from other sources in the figure text.

The figure also raises some other questions less relevant to your objective, e.g., why SSP3-70 bends and starts tracking downwards immediately after 2100 (that's rather strange, or maybe not according to Meinshausen or your assumptions) ... ecological breakdown, mass-extension of species, ... not mitigation, I assume. In essence, how does your model explain the SSP behaviors?

Table 2: Nice and informative

Line 124: “Even though the SSP emissions are constrained to reach zero in 2250,”

Which SSP emissions? Some scenarios reach zero and below before 2100

Again, I am curious about the assumptions or techniques that define the SSP emissions post-2100

Line 130: Median or mean? The measures in the text and Figure 2 are different. It's okay; just make it clear that you discuss both and that it is not a slip.

Line 141+: I am curious why you present the full uncertainty ranges. (I understand it better after reading the discussion, but at first, I was curious.)

I would include emissions in the figure text.

Discussion

Lines 181-195: It would be interesting to discuss or justify why 8.5 and 7.0 track down quite fast post-2100. These do not fit very well with the SSPs' pre-2100 assumptions. A clarification or at least a discussion of this would be good... at least to show that you are aware.

260-267: Nice you include this - Relevant since no one actually knows the future. Here, you could also discuss the reliability of your global SSP extensions post 2100 and if SSP3-7.0 and SSP5-8.5 really track downward this fast and steadily (and how this could happen in the real world - which factors could explain this/is it realistic)

267-276. Summary: Different methods exist for creating future emission scenarios. The RFF-SPs and Liu and Raftery (2021) approaches offer contrasting results, particularly in the median and lower emission ranges. The complexity of Morris et al. (2022) approach (including other variables (policy) makes direct comparison with RFF-SP difficult.

Does this mean that RFF-SPs do not include any socio-economic variables?

276-: Again, I miss some clarity about your aim with introducing the BAU and best estimate. It is great that you address the best guess concept, but it is not clear why and whether you operate with one or not. And why/why not?

I miss a conclusion that puts your aim and results into context and clearly summarizes the long discussion/states your position/recommendation. Not all readers will take the time to read the rather long discussion.

From lines 283 to 328, I skimmed the text.

I did not read the methods from A to Z, but some passages. I could not see anything about why you chose the Rennert scenarios. It is relevant to know why you chose these in particular. What makes it a good choice/less good choice (e.g., what do they overlook, miss, not include, e.g., policy (Morris))?

And your model, how does it explain the SSP behaviors post 2100?

REVIEWER COMMENTS

Reviewer #1 (Remarks to the Author):

The authors have attempted to address the comments raised in the previous round. By sending the paper back out for review, the editors seem to have made a decision about the paper's novelty being sufficient for publication. If that is indeed the case, I suggest providing the authors with one more chance to revise. In their revision, I encourage them to consider the following small but important points:

1. Probability: I still think the framing of the results as probabilistic deserves at least a caveat at the beginning, in addition to the one added to discussion. The RFF framework does not capture all possible sources of uncertainty, even though it tries to capture the range of expert opinion about uncertainty. These two things have sometimes been drastically different from each other, for example in the years leading up to the 2008 financial crash, or in advance of the COVID-19 pandemic.

Note that the use of the RFF-SPs in this paper is to produce global emission trajectories. Historically, short-term events (such as the 2008 financial crash and the 2020 COVID pandemic) which have large implications for society have only caused small deviations in the long-term carbon emission trend (see, e.g., <https://explore.globalcarbonbudgetdata.org/>) and so are not necessarily relevant for long-term projections like the RFF-SPs. The more important uncertainties are the long-term drivers such as the rate of economic growth and technological progress, and political constraints on emissions. But the purpose of an expert driven process like the RFF-SPs is to produce probabilistic estimates of these future possibilities.

The RFF-SPs are of course, not perfect. We discuss this at length in the discussion starting at line 260. We also believe that the general user of projections knows that there is no such thing as a perfect crystal ball: as the saying goes (sometimes attributed to Niels Bohr, sometimes to Yogi Berra), "It is difficult to make predictions, especially about the future."

We have added a statement at the end of the introduction: "We recognize the challenges of accurately capturing the probability of rare events that might have substantial consequences for future emissions and concentration changes but quantitative analysis of these probabilities has value despite the underlying limitations."

2. In Table 1, the exceedance probability is sometimes the same or higher for SSP4-3.4 as it is for SSP1-2.6. I think this is because the emissions trajectory for SSP4-3.4 dips slightly below SSP1-2.6 right before 2100, but to readers who don't know the SSPs well, this will seem like a very strange result (because 4-3.4 is a warmer scenario than 1-2.6). Therefore, I suggest explaining it.

One of the challenges of interpreting SSP and RCP scenarios is many of the scenarios come from different worlds produced by a different integrated assessment model, rather than a single world on which different emissions constraints have been imposed. For this reason, RCP4.5 CO₂ emissions exceed RCP6.0 emissions from 2005 through 2050 (see

<https://tntcat.iiasa.ac.at/RcpDb/dsd?Action=htmlpage&page=compare>), and SSP5-45 reaches 574 ppm of CO₂ in 2100 whereas SSP4-45 only reaches 512 ppm (see <https://tntcat.iiasa.ac.at/SspDb/dsd?Action=htmlpage&page=40>). There are even two different SSP3-7.0s, one with high pollutant emissions and one with low pollutant emissions. (See also <https://www.carbonbrief.org/explainer-how-shared-socioeconomic-pathways-explore-future-climate-change/>). See also Moss et al. (2010) (Moss, R., Edmonds, J., Hibbard, K. et al. The next generation of scenarios for climate change research and assessment. Nature 463, 747–756 <https://doi.org/10.1038/nature08823>) where they state that the scenarios cannot “be treated as a set with consistent internal logic.”

We have added a sentence before Figure 1/Table 1 explaining this phenomena: “The fact that the SSP1 family of scenarios and the SSP4 family of scenarios are developed by different modeling groups with different assumptions about underlying economic and technological drivers explains how the emissions from the higher SSP4-3.4 scenario can drop below the emissions from the lower SSP1-1.9 and SSP1-2.6 scenarios by 2100, recognizing that the latter two scenarios still have a lower radiative forcing in 2100.”

Reviewer #3 (Remarks to the Author):

The paper now reads better after clarifying many issues raised by me and other reviewers. My comments in this round are as follows.

Thank you, we agree that the paper reads better after considering your suggestions.

1. The authors state that 53 of the 10,000 RFF-SP scenarios exceed 8.5 W/m² in 2100, showing that the probability of radiative forcing exceedance can be substantially larger than the probability of exceeding the emissions considered in earlier literature. What causes the difference between the probability of radiative forcing exceedance and emissions exceedance? Can you elaborate on this point and give some explanations in more detail?

Our explanation was as follows: “In this case, the probability of exceeding the emissions in the SSP5-8.5 scenario in 2100 when taking the RFF-SP distribution as truth is 3 in 10,000. Even the probability of exceeding SSP3-7.0 emissions by 2100 is on the order of 1%. However, when we move to radiative forcing, the probabilities become substantially larger: 0.5% for exceeding 8.5 W m⁻², and 7% for exceeding 7 W m⁻². A key contributor to this increase in the probability of exceedance is the inclusion of additional uncertainties, in particular carbon cycle and methane feedbacks.” We’ve added a final sentence to this paragraph: “As a first order effect, adding an additional uncertainty to an existing probability distribution will lead to an increase in the size of the tails of the distribution.”

This kind of result was anticipated by studies such as Hausfather & Peters (2020) which showed that SSP5-8.5 was extreme in emissions space, but recognized that lower emissions scenarios could result in radiative forcing and temperature futures similar to SSP5-8.5 due to these added uncertainties such as in the carbon cycle and climate sensitivity.

2. The uncertainty part is crucial to the results but the authors didn’t positively address much of my concerns in my previous comment #4. My further question is from your new Fig.3, can we conclude that

emissions uncertainty is the largest source of uncertainty to climate model projections as greenhouse warming continues?

The discussion paragraph regarding Figure 3 has been rewritten for clarity, and a statement about emissions uncertainty becoming the largest source of temperature uncertainty after 2100 has been added:

Over a timescale of a few decades, climate parameter uncertainty is the dominant driver of uncertainty in radiative forcing as emissions pathways have yet to diverge substantially due to economic inertia whereas the rate of carbon uptake and methane atmospheric chemistry both contribute to uncertainty in future concentrations (Figure 3a and b). However, by mid-century, emissions uncertainty becomes the dominant driver of radiative uncertainty as by 2050, the 95th percentile of CO₂ emissions is 6 times the 5th percentile. In contrast, for temperature, internal variability is initially a substantial component of temperature uncertainty, becoming a smaller proportion of the total over time despite remaining relatively constant in absolute terms (as in Hawkins & Sutton (2009)). Year-to-year variability in temperature due to factors such as ocean currents are of the same magnitude (about 0.2 °C) as the recent decadal trend in temperature. Meanwhile, climate uncertainty continues to be a larger contributor than emission uncertainty through the end of the century (Figure 3c and d). This is a result of fewer uncertain climate parameters that impact radiative forcing (e.g. aerosol forcing, carbon cycle, and methane feedback uncertainties) than impact temperature (e.g. all the parameters that impact radiative forcing also matter for temperature, but uncertainties in climate sensitivity and ocean heat uptake also have substantial temperature impacts). Due to the cumulative nature of carbon in the atmosphere, the contribution of emissions uncertainty continues to grow with time, becoming the most important factor for understanding temperature trends on the multi-century scale. While most of this paper discusses radiative forcing as it is the key input to climate models, temperature uncertainty is important as temperature drives impacts that are relevant to human and ecological experiences. Two key advances over the previous Hawkins and Sutton (2009) and Lehner et al. (2020) are the use of fully probabilistic emission and climate parameter uncertainty inputs for this analysis (rather than, for example, assuming that the RCPs are all equally likely, or that all climate models are equally likely, as was done for the previous analyses).

3. As the core of this study is based on the simple climate model FaIR, you'd better provide some evidence showing that the model is reliable and good at capturing observations, especially to people outside of the field and not familiar with the model and methods.

Added a statement, "The FaIR model is a widely used reduced complexity model that has been relied on by major institutions such as the IPCC (IPCC, 2021), the US National Academies of Science (NAS, 2017), and the EPA Social Cost of Carbon (EPA, 2023), and which has participated and performed well in the RCMIP intercomparison project (Nicholls et al. 2020, 2021)." The model calibration process ensures that the model is consistent with historical data and understanding of large scale climate processes: "The parameters for the model were calibrated to be consistent with historical data taking into account

observational uncertainty (ocean heat content, CO₂ concentrations, aerosol radiative forcing) as well as consistency with the ranges for transient climate response and equilibrium climate sensitivity presented in the IPCC AR6 Working Group 1 assessment (Forster et al. 2021; Smith et al. 2021). Each member of the ensemble maintains a root-mean-square error between modelled and observed global mean surface temperature of less than 0.16°C...”

A paper devoted to the calibration and constraining process in FaIR has been submitted to Geoscientific Model Development and is now available as a discussion preprint at <https://egusphere.copernicus.org/preprints/2024/egusphere-2024-708/>. Also, a demonstration of this in published literature exists in Smith et al. 2023, <https://iopscience.iop.org/article/10.1088/1748-9326/acdc6>, particularly the supplementary material of that paper.

4. Fig.1: the colors used for now are extremely difficult to distinguish different SSPs

Colors have been updated, and the lines have been made thicker to make it easier to distinguish between scenarios.

5. Line106: should be “SSP3-7.0”

Corrected, thank you.

6. Line148: with an 88% chance of exceeding 2oC in 2100?

Added, “in 2100” – thank you: “The median temperature in 2100 from the RFF-SP/FaIR ensemble is 2.9 °C (5th to 95th percentiles from 1.7 to 4.2 °C), with an 88% chance of exceeding 2 °C.”

7. Line377: what does the 50th percentile mean?

Thank you, we agree this wasn’t explicit. Sentence updated to: “We use the 50th percentile of emissions across all scenarios in the AR6 infiller database for each short-lived forcer”.

This method uses a quantile regression of emissions of the species you want to obtain but don’t have in your scenario (e.g. SO₂), using a database of existing integrated assessment model derived scenarios (in this case from IPCC AR6 WG3), to an emissions species that you do have (e.g. CO₂). The “rolling windows” part of this method performs the quantile regression over a moving window of time points, noting that these relationships can change over time (such as expected future strengthening air pollution clean up due to air quality legislation, even in scenarios with high CO₂ emissions). Infilling using QRW is an established technique and used extensively in the IPCC AR6 (Kikstra et al. 2022, <https://gmd.copernicus.org/articles/15/9075/2022/>)

8. As acknowledged by the authors that individual uncertainty components may exceed the total uncertainty, the use of limit to 100% in y-axis in Fig.3 is not appropriate and misleading.

We updated the methodology for producing Figure 3, such that we now divide the total uncertainty into the three categories in proportion to the individual uncertainty of each. Therefore, by definition, this method now yields a total contribution of 100%.

Reviewer #4 (Remarks to the Author):

Nice improvements. Well done. Reads better now. I like the paper and enjoyed reading it. It is almost done, in my opinion, but there are still some issues to solve :)

Thank you for your suggestions, we agree that they improved the paper.

I made several repetitions - I didn't delete them all, so reply once to each point ;)

I like the introduction better, and it focuses on adding probability to the scenarios. Just remember that the RCP-SSP scenarios were never intended to be based on frequency distributions or probability analysis—they are all plausible scenarios without probability attached. So it is actually great that you question why the RCP modelers added probability to 8.5 as the only scenario—maybe explain better how they should have done it, in your opinion.

This paper presents one approach to developing probabilities for radiative forcing exceedances. The challenge for the RCP/SSP scenario developers is that GCMs are very computationally expensive, so the number of key scenarios has to be limited to a handful – and it is hard to be rigorous about probabilities with only a handful of scenarios. For example, we find a 42% chance of emissions exceeding the 21.9 GtCO₂ from SSP4-6.0 in 2100 (Table 1), but there are many ways of reaching 21.9 GtCO₂ in that year. It could be a peak and decline scenario, or a slow growth and then acceleration scenario, or a constant moderate growth scenario. With 10,000 RFF-SP scenarios, all of these options can be included in proportion to their likelihood, but with a half-dozen scenarios to cover all possible futures, only one (or in some cases, two) scenarios can be chosen for any given radiative forcing endpoint.

We have added a paragraph explaining this difference:

This kind of probabilistic analysis is made possible by the use of thousands of probabilistic economic scenarios coupled with thousands of probabilistic parameter sets from a reduced complexity climate model. Given the computational demands of full complexity GCMs, the scenario community has generally chosen fewer than a dozen primary scenarios to drive these GCMs. A proper probabilistic analysis would be impossible with so few emissions or concentration scenarios, and then the GCMs themselves are not designed for probabilistic analysis. Therefore, the kind of probabilistic analysis highlighted in this paper is designed as a complement to the more traditional scenario-based approach which produces a finite set of alternative future storylines without assigned probabilities.

Make it clear in the introduction or results that you compare two different types of scenarios. It could be something like the following:

The RFF-SPs are probabilistic emission projections, forecasts about future greenhouse gas emissions (and radiative forcings) that account for uncertainty, e.g., presenting a single (best-guess??) prediction including a range of possible outcomes or 82 probabilistic emission predictions. Forecasts often include predicting likely futures based on frequency distributions. These are compared to the RCP-SSPs and RCPs scenarios. The RCP-SSPs are different from the RFF-SPs. They are qualitative descriptions of alternative futures. They are not forecasts or predictions of most likely futures but descriptions (or projections) of alternative futures that are plausible but not necessarily probable. They are typically based on different sets of assumptions about how various factors might evolve over time. Scenarios are used to explore the range of possible outcomes and to understand the potential consequences of different decisions or events (e.g., Webster et al., 2003; Pedersen et al., 2022).

See previous response: we have added a paragraph discussing the differences between large ensemble probabilistic approaches vs. small groups of plausible future scenario approaches.

In the abstract, I would state that the low end of the SSP-RCP range is within the range of your results, while SSP5-8.5 is outside the upper end. That way, your results are presented in a more complete and balanced way, and it highlights your aim: to discuss SSP5-8.5.

We have added the probability of being lower than SSP1-2.6 to the abstract.

Focusing on (the low probability of) RCP8.5 is relevant; however, it has always been a low-probability, high-impact case aiming to inform the worst possible outcome for adaptation analysis (as you also more or less address). I would be surprised if it will be excluded from the upcoming SSP elaborations or CIMP for the AR7 analysis. I think your introduction puts it much more to the point with the statement from the ScenarioMIP workshop report.

The ScenarioMIP workshop report sentence is in line with some other statements I've heard from some modelers in the community, which was part of the driver for writing this paper.

I'm curious about your model and the probabilistic analysis beyond 2100. On which basis did you develop the SSP extensions beyond 2100, e.g., what are the underlying assumptions (e.g., mathematical, socioeconomic)? In particular, they are relevant to discuss because 7.0 and 8.5 start tracking downward quite fast after 2100. Why? This info is missing in the methods (I think?). You will demonstrate a good understanding and critical self-evaluation if you elaborate on why you track down this fast in the discussion section and also discuss the reliability of tracking downward at a relatively fast speed.

We have a paragraph in our methods discussing the post-2100 extrapolations:

We follow the logic discussed in Meinshausen et al. (2020) to extend SSP emissions beyond 2100. If CO₂ FFI emissions are positive in 2100, they are linearly ramped down to zero by 2250; if

they are negative, they are ramped up to zero by 2150. CO₂ AFOLU emissions are ramped to zero by 2150 whether negative or positive in 2100.

This Meinshausen approach (<https://doi.org/10.5194/gmd-13-3571-2020>) is the standard post-2100 extrapolation, as demonstrated by the 750 or so citations that paper has received. I agree that this extrapolation is very stylized, but there are no agreed upon alternatives (you will be pleased to know there is a dedicated team working on how best to extend scenarios beyond 2100, or likely 2125 which is proposed as the time horizon of IAMs, for CMIP7). Also, our quantitative comparisons in the tables are related to the 2100 numbers from the SSPs, so it is only the graphs where the post-2100 assumptions become relevant.

Does your “mean” estimate represent a best-guess scenario? I’m curious. You mention the concept but don't say whether you are presenting one. Between RF 4.5 and 6.0 is not far from most assessments of current policies and/or NDCs.

The mean represents the average emissions / radiative forcing / temperature at any given point in time. There will not be any single scenario which actually follows the mean pathway over time. An alternate approach would be to pick the mean/median scenario in a given year (e.g., 2100) and then show its pathways over time – but the problem with that approach is that the scenarios just above and just below that scenario in 2100 could have very different pathways over time, so that would present some arbitrariness. (Again, this a difference between probabilistic approaches and scenario based approaches).

To sum up, the most important issues to resolve and clarify: - why does the paper discuss BAU and best-estimate (intro and discussion); and if it presents a best-estimate scenario or not (and why or why not if it does not present one)

In the abstract, we state that “The median forcing in 2100 resulting from this probabilistic analysis is 5.1 W m⁻² (95th percentiles, 3.3 to 7.1 W m⁻²)”. This could be a guide for creating or choosing a BAU scenario. However, as noted above, because the 5.1 W/m² scenario might look very different from the 5.09 and the 5.11 W/m² scenarios (even at the global scale – at the regional scale, even more so!), calling any one scenario from the probabilistic ensemble a “best estimate” or “BAU” would not be appropriate. This is why this kind of approach is a good complement to the storyline approach. We have also removed several of the references to BAU in the paper.

- Why SSP5-8.5 and SSP3-7.0 track downward rather fast beyond 2100 - some mathematical assumptions and a critical discussion about the reliability in a real-life world of this to happen. I’m not saying it is wrong, but I would expect some critical self-reflections - it would be interesting and valuable with a reliability discussion both within the model and its inputs and between the model and the real world. In essence, how do your model or assumptions explain the SSP behaviors post-2100?

See above response - This Meinshausen approach (<https://doi.org/10.5194/gmd-13-3571-2020>) is the standard post-2100 extrapolation, as demonstrated by the 750 or so citations that paper has received. I agree that this extrapolation is very stylized, but there are no agreed upon alternatives. Also, our

quantitative comparisons in the tables are related to the 2100 numbers from the SSPs, so it is only the graphs where the post-2100 assumptions become relevant.

(note that for SSP5-8.5 in particular, limitations in fossil resources would likely kick in requiring emissions decreases. Moreover, some Earth Systems Models might crash at the very high concentrations and temperatures resulting from continued extremely high emissions post 2100)

- Clearly state the differences between the scenarios you compare, how are they comparable, and the conditions for comparing to different (sets and) types of scenarios. The RFF-SR are probabilistic emission projections, while the SSP-RCPs are not. It's fine to compare them, but it is essential to state the differences and the conditions for comparisons (introduction or results and discussion, and maybe methods).

See the added paragraph within the discussion section.

- A conclusion where you summarize the results and long discussion and clearly state your recommendation (opinion) - not everyone will read the entire discussion.

See the addition of the "in summary" statement before the third to last paragraph of the discussion: these last three paragraphs can be considered our summary/conclusion with recommendation.

- The modelers never intended the BAU concept. It was labeled like that by several scenario users. It is ok to discuss it but don't make the same mistakes as others... in particular if it is part of your justification to examine 8.5 - check I think Pedersen et al. 2021 or 2022 (one of these presents the historical use of BAU since the first IPCC scenarios in 1990 and how modelers started to use a DAU concept instead - but apparently the users continued with BAU :) - the 2021 paper looks at scenarios vs. history and must be easy to find.

Revisiting the one mention of BAU in this paper, I decided that it was actually irrelevant to the point I was making, and I have deleted that mention.

I also see that dynamics-as-usual "DAU" was used at least as far back as 2000 with Riahi et al. (<https://www.sciencedirect.com/science/article/pii/S0040162599001110#BIB8>) – I had not been previously familiar with the term, thank you for introducing me to it.

Why did you choose the Rennert scenarios? e.g. What are the advantages? (In "Methods," add a line in the nice paragraph explaining the scenarios in the "introduction.")

Our primary justification for the use of the RFF-SP scenarios is presented in the following sentence on line 351: "RFF-SP is recent, open-source, fully probabilistic, extends through the year 2300, and is used by the U.S. SC-GHG process, making it an ideal dataset for emission projections" (though our discussion comparing the RFF-SPs to Liu and Raftery and Morris et al. is also relevant). We are not aware of any other recent probabilistic emission scenarios that are internally self-consistent and extend through 2300: Morris et al. and Raftery & Liu only projected emissions through 2100.

Regarding the SSP behavior post-2100: the standard approach to extending the SSPs is to use the Meinshausen et al. (2020) extension method – see the paragraph starting on line 393 describing these assumptions (which include ramping fossil fuel CO2 emissions to zero by 2250).

Abstract:

It's quite improved and nice, with the final putting the results into perspective. However, it's still quite technical, which narrows your reader group. I would prefer you begin with 1-2 sentences comprising some context and introducing the field and problem you wish to address.

Less technical language could invite readers who are normally far from or outside the statistical field to understand how our results are useful. For example, damage function could be swabbed with or supplemented by impact or adaptation analysis. This may bring your writing closer to the real world and maybe also inform decision-making or disaster-planning actors.

Added a new start to the abstract: “Future climate projections are often future storylines without assigned probabilities.” Because of the 150 word constraint, we were not able to provide more substantial context, unfortunately.

Introduction:

Very improved introduction. I think you will catch your audience much better here and introduce your aim well.

Thank you for your helpful suggestions.

Line 50 (+ Discussion 276): I would modify this. RCP-85 was never intended to be a BAU; however, it was used by several scenario users as such. This was explained in one of the Pedersen et al. papers, maybe in 2021 or 2022 (IPCC mandated a stop of this term, so modelers started using dynamics as usual instead). You could write that it was mistakenly perceived as a BAU by some scientific scenario users but was never intended as such by the developers. As I remember, Hausfather/Peters (2020) misinterpreted it a bit - which happens, but don't do the same ;)

The reference to BAU has been deleted.

Line 51: I'm not entirely certain that your RFF-SP represents a best-estimate scenario. Could you clarify whether it does or does not?

The RFF-SPs are designed to represent a probabilistic projection of the future: the 10,000 scenarios together could be considered a “best-estimate” of the range of possibilities, but no single scenario within would be.

Lines 80-90. I like the inclusion of this paragraph—it supports reading the results and can be improved further, E.g., clarify what variables the analyses are based on. Is the statistical analysis based on emissions and forcings alone? Or does it also include other variables like economy, policy, and

population (which are included in the SSPs and SPAs (and SSP-RCPs))?

You could additionally write in a few sentences clarifying for non-statisticians what the probability assessment covers in a non-technical language.

In essence, I would improve the quite nice and short description (lines 80-90), opening up for a better understanding of the rather abstract and less transparent functions. It is distracting to read the results when I am unsure about what the probability is based on (it would be nice to know what is lying behind)

We have expanded the sentence introducing the RFF-SPs to detail the underlying process as follows:

First, we use a recently developed ensemble of probabilistic emission projections based on expert elicitation of future rates of change of country-level population growth, economic growth, and emissions intensity (the Resources for the Future Socioeconomic Projections, or RFF-SPs, Rennert et al. 2022a, Rennert et al. 2022b)

It could be something like:

The RFF-SR are probabilistic emission projections or forecasts about future greenhouse gas emissions that present a single prediction accounting for uncertainty or ... instead of a single prediction, they provide a range of possible outcomes. They are based on the Future Socioeconomic 82 Projections (RFF-SPs), developed by Rennert et al. (2022ab), based on ...

(Just ideas, use your own words to make it transparent what variables the analysis is based on, Kyoto GHGs or also other variables - I have not read the methods yet). After reading the entire paper, I would suggest you summarize the main characteristics of the Rennert scenarios and your model. Make it more transparent what your scenarios represent - they are more than numbers and equations, open them up for the reader so we can better evaluate/interpret the results

Line 39-40: "Shared Socioeconomic Pathway (SSP) 5-8.5 for CMIP6". I would write the RCP-SSP combination and include Gidden et al. (2019)

We have added the reference to Gidden et al. (2019) and updated the text around SSP-RCP.

Lines 54-56: references seem to be missing

Added references to Gidden & Riahi here.

Lines 86-87: Are any references needed here?

This is a description of the calculations done in this paper, no references needed

Line 88: "and other dates". Dates or years? Why not write the exact years?

Good point: explicitly mentioned 2150 and 2300 as the other years involved.

88 -90: Belongs to results

I recognize what you are saying, but we see value to including one or two “we show” sentences to preview the rest of the paper in the introduction.

Results:

Interesting results, both emissions and RF.

The findings would appear more trustworthy for non-statisticians if I understand better what the RFF-SP probability expresses or is based on (i.e., include in the description in lines 80-90)

See above addition:

First, we use a recently developed ensemble of probabilistic emission projections based on expert elicitation of future rates of change of country-level population growth, economic growth, and emissions intensity (the Resources for the Future Socioeconomic Projections, or RFF-SPs, Rennert et al. 2022a, Rennert et al. 2022b)

Fig. 1: Nice figure btw :)

Thank you

What data source did you use for the RCP-SSPs in Fig. 1? If you didn't generate the SSP data yourselves, I would add that source in the figure text (for the SSP database, Riahi et al., 2017 will be sufficient).

NB: After reading the methods, I see that you extended the SSPs. Please clarify what data you used and what is from other sources in the figure text.

Added reference to Meinshausen et al. (2020) in the figure caption

The figure also raises some other questions less relevant to your objective, e.g., why SSP3-70 bends and starts tracking downwards immediately after 2100 (that's rather strange, or maybe not according to Meinshausen or your assumptions) ... ecological breakdown, mass-extinction of species, ... not mitigation, I assume. In essence, how does your model explain the SSP behaviors?

The Meinshausen SSP extensions were designed to be stylistic. While most of the analysis in this paper only relies on SSP data in the year 2100, we felt it was still relevant to compare the full time-line of RFF-SP projections to the most-used SSP extension (e.g., Meinshausen). Added “The Meinshausen et al. (2020) approach to extending the SSPs uses very stylized assumptions about emissions converging to zero by 2250” in the discussion of extensions through 2300.

Table 2: Nice and informative

Thank you

Line 124: "Even though the SSP emissions are constrained to reach zero in 2250,"

Which SSP emissions? Some scenarios reach zero and below before 2100

Again, I am curious about the assumptions or techniques that define the SSP emissions post-2100

Added the detail "fossil fuel emissions" for the constraint to reach zero: note that the constraint also applies to negative emissions that are constrained to increase until they reach zero. See description of the Meinshausen technique (original paper, lines 393-400). Added "The Meinshausen et al. (2020) approach to extending the SSPs uses very stylized assumptions about emissions converging to zero by 2250" in the discussion of extensions through 2300.

Line 130: Median or mean? The measures in the text and Figure 2 are different. It's okay; just make it clear that you discuss both and that it is not a slip.

Yes: the text in line 130 refers to the median, because it is about likelihood and probabilistic bounds, but the figure shows the mean because that is often the key metric for impacts because it reflects that the underlying distribution might be skewed.

Line 141+: I am curious why you present the full uncertainty ranges. (I understand it better after reading the discussion, but at first, I was curious.)

I would include emissions in the figure text.

We do mention emissions in the figure caption: "showing the 5th to 95th percentiles from emissions uncertainty, climate parameter uncertainty, internal climate variability, and total uncertainty in (a) and (c)"

Discussion

Lines 181-195: It would be interesting to discuss or justify why 8.5 and 7.0 track down quite fast post-2100. These do not fit very well with the SSPs' pre-2100 assumptions. A clarification or at least a discussion of this would be good... at least to show that you are aware.

Again, the Meinshausen et al. future projections are very stylistic: the emissions are constrained to reach zero in 2250, without any mechanistic reasoning behind that assumption. Because Meinshausen is the most-used post-2100 projection for the SSPs, it is reasonable to rely on this approach. Added "The Meinshausen et al. (2020) approach to extending the SSPs uses very stylized assumptions about emissions converging to zero by 2250" in the discussion of extensions through 2300.

260-267: Nice you include this - Relevant since no one actually knows the future. Here, you could also discuss the reliability of your global SSP extensions post 2100 and if SSP3-7.0 and SSP5-8.5 really track downward this fast and steadily (and how this could happen in the real world - which factors could explain this/is it realistic)

Added the following sentence: “The Meinshausen et al. (2020) approach to extending the SSPs uses very stylized assumptions about emissions converging to zero by 2250.”

267-276. Summary: Different methods exist for creating future emission scenarios. The RFF-SPs and Liu and Raftery (2021) approaches offer contrasting results, particularly in the median and lower emission ranges. The complexity of Morris et al. (2022) approach (including other variables (policy) makes direct comparison with RFF-SP difficult.

Does this mean that RFF-SPs do not include any socio-economic variables?

The reason that Morris et al. (2022) can't be directly compared to the RFF-SPs is not the complexity of the approach (though Morris et al. does rely on a more complex model), but rather that Morris et al. produce four different probabilistic distributions for four different possible futures. Therefore it is unclear what combination of these four futures should be compared to the RFF-SPs. We edited the final sentence of the Morris et al. description to state,

“However, Morris et al. separate policy uncertainty out as a separate factor with four possible futures, making it difficult to compare directly to the RFF-SP approach which treats policy probabilistically.”

276-: Again, I miss some clarity about your aim with introducing the BAU and best estimate. It is great that you address the best guess concept, but it is not clear why and whether you operate with one or not. And why/why not?

Modified sentence to state, “Addressing the potential for future policy in a probabilistic framework is challenging” and remove the reference to “business as usual”.

I miss a conclusion that puts your aim and results into context and clearly summarizes the long discussion/states your position/recommendation. Not all readers will take the time to read the rather long discussion.

Nature Communications research articles do not generally have a separate conclusion section. To address your comment, we have added the phrase “in summary” to the third to last paragraph, as those last three paragraphs can be considered equivalent to a conclusion section.

From lines 283 to 328, I skimmed the text.

I did not read the methods from A to Z, but some passages. I could not see anything about why you

chose the Rennert scenarios. It is relevant to know why you chose these in particular. What makes it a good choice/less good choice (e.g., what do they overlook, miss, not include, e.g., policy (Morris))? And your model, how does it explain the SSP behaviors post 2100?

Our primary justification for the use of the RFF-SP scenarios is presented in the following sentence on line 351: “RFF-SP is recent, open-source, fully probabilistic, extends through the year 2300, and is used by the U.S. SC-GHG process, making it an ideal dataset for emission projections” (though our discussion regarding Raftery & Liu and Morris et al. is also relevant). We are not aware of any other recent probabilistic emission scenario that extends through 2300: Morris et al. and Raftery & Liu only projected emissions through 2100.

Regarding the SSP behavior post-2100: the standard approach to extending the SSPs is to use the Meinshausen et al. (2020) extension method – see the paragraph starting on line 393 describing these assumptions (which include ramping fossil fuel CO2 emissions to zero by 2250).

REVIEWER COMMENTS

Reviewer #1 (Remarks to the Author):

Although I think the authors and I still have a difference of opinion regarding the ability of multi-century models and expert opinion to capture uncertainty, I am satisfied with the caveats they have added to the paper.

Reviewer #3 (Remarks to the Author):

The authors have successfully addressed my concerns and the manuscript has significantly improved. I do not have any additional comments.

Reviewer #4 (Remarks to the Author):

Improved paper with interesting results for a broader and a more narrow scientific community. First, concerning the assessment of the median forcing in 2100 of 5.1 W m^{-2} (5th to 95th percentiles of 3.3 to 7.1 W m^{-2} 30) and probabilities of 8.5 W m^{-2} , and a 1% probability of being lower than 2.6 W m^{-2} . second, the uncertainty and relevance of variables and probability. The quite dense discussion focuses on radiative forcing as the best metric for representing climate impact, though it has limitations and the importance of using probabilistic approaches for climate scenario analysis.

Lines 47-51: I don't understand this sentence. Try to break it up into smaller sentences with one message in each. Furthermore, best-guess scenarios would probably not be the majority but one or a few of a series of scenarios.

“ [...] this 10% statistic may bear little resemblance to an actual probability of exceedance: the majority of the scenarios are likely to be designed as “best-estimates” rather than being chosen to span a full distribution of possibilities, thereby potentially underweights the tails; and carbon cycle uncertainties may not have been included in all the studies, again likely underweighting tails.”

Lines 76-78: I would like you to explain better the relevance of RQ2, rephrase it, and explain how you would analyze it. RQ 1 is straightforward. Do you need RQ2? I have difficulties understanding what

you wish to answer, why, and how. What do you mean by appropriate likelihood ... the importance of choosing the right way/variable to express the likelihood of scenarios? Is it a statistic analysis or rather a discussion, reflecting on/comparing "CO2" with "forcing probability density function"?

I suggest rephrasing RQ2 and explaining how you will approach it in a more straightforward way (actually, I'm not sure that you explain RQ2 below). Try to consult some non-statistic researchers. The term "Appropriate likelihood" is not mentioned in the results or discussion.

Lines 166-366: Long section. I suggest some sub-sections in the discussion to support readability.

As I read the dense section, it could be structured via some of the below headlines/content:

- Probability of high-forcing scenarios declining: The likelihood of exceeding a specific forcing level (8.5 W/m^2) has decreased over time due to lower emission projections.
- 8.5 W/m^2 scenario remains relevant. Despite its low probability of occurring by 2100, this scenario is useful for various applications, like creating damage functions for climate impacts, simulating future climates beyond 2100 using "analog" approaches and informing policy decisions regarding low-probability, high-impact events.
- Future considerations: If impact analyses extend beyond 2100, a high-end scenario reaching 8.5 W/m^2 in a later year might be more relevant.
- Hope for continued emission reduction: As mitigation efforts increase, the likelihood of extreme scenarios may decrease further.

Finally, I suggest a concluding section at the end to catch more readers of the article (and increase citations of your paper)

REVIEWER COMMENTS

Reviewer #1 (Remarks to the Author):

Although I think the authors and I still have a difference of opinion regarding the ability of multi-century models and expert opinion to capture uncertainty, I am satisfied with the caveats they have added to the paper.

Thank you – we recognize that projecting emissions 300 years into the future is an exercise in assumptions, but it is also an important input for developing policy-relevant metrics such as the social cost of greenhouse gases. We appreciate your input regarding appropriate caveats to include.

Reviewer #3 (Remarks to the Author):

The authors have successfully addressed my concerns and the manuscript has significantly improved. I do not have any additional comments.

Thank you again for your productive suggestions for improving the manuscript.

Reviewer #4 (Remarks to the Author):

Improved paper with interesting results for a broader and a more narrow scientific community. First, concerning the assessment of the median forcing in 2100 of 5.1 W m^{-2} (5th to 95th percentiles of 3.3 to 7.1 W m^{-2}) and probabilities of 8.5 W m^{-2} , and a 1% probability of being lower than 2.6 W m^{-2} . second, the uncertainty and relevance of variables and probability. The quite dense discussion focuses on radiative forcing as the best metric for representing climate impact, though it has limitations and the importance of using probabilistic approaches for climate scenario analysis.

Lines 47-51: I don't understand this sentence. Try to break it up into smaller sentences with one message in each. Furthermore, best-guess scenarios would probably not be the majority but one or a few of a series of scenarios.

" [...] this 10% statistic may bear little resemblance to an actual probability of exceedance: the majority of the scenarios are likely to be designed as "best-estimates" rather than being chosen to span a full distribution of possibilities, thereby potentially underweights the tails; and carbon cycle uncertainties may not have been included in all the studies, again likely underweighting tails."

Thank you for the comment. We have decided to simplify the sentence as follows: "Because the underlying scenarios were not designed to be equally likely, the fact that 8.5 W/m^2 represented the 10th percentile of available scenarios does not imply an assessment that there was a 10% likelihood of exceeding 8.5 W/m^2 ", as the discussion of whether 10% was more likely an overestimate or underestimate was unnecessary.

Lines 76-78: I would like you to explain better the relevance of RQ2, rephrase it, and explain how you would analyze it. RQ 1 is straightforward. Do you need RQ2? I have difficulties understanding what you

wish to answer, why, and how. What do you mean by appropriate likelihood ... the importance of choosing the right way/variable to express the likelihood of scenarios? Is it a statistic analysis or rather a discussion, reflecting on/comparing "CO2" with "forcing probability density function"?

I suggest rephrasing RQ2 and explaining how you will approach it in a more straightforward way (actually, I'm not sure that you explain RQ2 below). Try to consult some non-statistic researchers. The term "Appropriate likelihood" is not mentioned in the results or discussion.

Thank you for the comment. We rephrase RQ2 as "and 2) what is the appropriate forcing to use for a high-end scenario in climate analysis?" which focuses on the forcing rather than the likelihood. The two are related, but most practitioners care about the former rather than the latter.

Lines 166-366: Long section. I suggest some sub-sections in the discussion to support readability. As I read the dense section, it could be structured via some of the below headlines/content:

- Probability of high-forcing scenarios declining: The likelihood of exceeding a specific forcing level (8.5 W/m²) has decreased over time due to lower emission projections.
- 8.5 W/m² scenario remains relevant. Despite its low probability of occurring by 2100, this scenario is useful for various applications, like creating damage functions for climate impacts, simulating future climates beyond 2100 using "analog" approaches and informing policy decisions regarding low-probability, high-impact events.
- Future considerations: If impact analyses extend beyond 2100, a high-end scenario reaching 8.5 W/m² in a later year might be more relevant.
 - Hope for continued emission reduction: As mitigation efforts increase, the likelihood of extreme scenarios may decrease further.

Finally, I suggest a concluding section at the end to catch more readers of the article (and increase citations of your paper)

We have added sub-sections, and done some minor reorganization:

- Applications of low-probability scenarios
- Characterizing exceedance probabilities
- Limitations
- Conclusions

Other reviewers had also suggested a concluding section: the format of Nature publications does not allow for a formal conclusion section, but the use of sub-sections allowed us to create one informally, so thank you for raising the sub-section idea.